# Unifying Agent Interaction and World Information for Multi-agent Coordination

## Abstract

This work presents a novel representation learning framework, *interaction-world latent* (`IWoL`), to facilitate *team coordination* in multi-agent reinforcement learning (MARL). Building effective representation for team coordination is a challenging problem, due to the intricate dynamics emerging from multi-agent interaction and incomplete information induced by local observations. Our key insight is to construct a learnable representation space that jointly captures inter-agent relations and task-specific world information by directly modeling communication protocols. This representation enables fully decentralized execution with implicit coordination while avoiding the drawbacks of explicit message passing, for example, slower decision-making, vulnerability to malicious attackers, and sensitivity to bandwidth limitations. In practice, our representation can be used not only as an implicit latent for each agent, but also as an explicit message for communication. Across four challenging MARL benchmarks, we evaluate both variants and show that `IWoL` provides a simple yet powerful key for team coordination. Moreover, we demonstrate that our representation can be combined with existing MARL algorithms to further enhance their performance.

## 1 Introduction

Representation learning has become a foundational paradigm, leading to significant advances in computer vision (Roh et al., 2021), natural language processing (Liu et al., 2023), and, more recently, learning-based control (Hafner et al., 2025; Espeholt et al., 2018). Within imitation learning and reinforcement learning (RL), structured latent variables such as successor features (Sun et al., 2025; Agarwal et al., 2024), temporal distance embedding (Wang et al., 2023; Ma et al., 2023), skill embedding (Pertsch et al., 2021; Wang et al., 2024c), and symbolic representation (Landajuela et al., 2021; Christoffersen et al., 2023) have improved sample efficiency and policy generalization. By contrast, effective representation learning in MARL remains relatively underexplored. Furthermore, partial observability and complex credit assignment blur the learning signal, hindering agents from acquiring representations for team coordination in their shared environment (Guan et al., 2022).

This work studies how to build a representation for a multi-agent system with the following question:

*What information should a latent representation for team coordination capture*
*to enable coordinated control from partial and noisy local observations?*

We argue that an effective representation for team coordination should capture at least two-fold: *(i)* inter-agent relations (Sukhbaatar et al., 2016; Foerster et al., 2016; Jiang & Lu, 2018), who influences whom, and how complementary their roles are, and *(ii)* task-specific world information, *i.e.*, a compact surrogate of the privileged global information (Cai et al., 2024; Li et al., 2024b; Ndousse et al., 2021; Salter et al., 2021). Such representation provides sufficient information for downstream control, permutation-invariant to agent ordering, and scalable with team size, so that decentralized policies can reason consistently about a shared environment despite non-stationarity and credit assignment challenges (Omidshafiei et al., 2017; Dibangoye & Buffet, 2018; Lee & Kwon, 2024a).

We address this question by learning an *interaction-world* representation through an encoder, trained with two decoders that align such a representation with the factors needed for coordination. More precisely, *interactive* and *world* decoders reconstruct pairwise interaction signals between agents,

which are extracted by a graph-attention mechanism (Veličković et al., 2018), and privileged state features, respectively. Note that two decoders and a graph-attention module are used only at training time as guidance; at execution, each agent conditions on its local observation to produce a unified representation without any decoder or message exchange, yielding a fully decentralized (and thus implicit) use of the learned representation. While the graph-attention approach resembles a communication module (Hu et al., 2024b; Pina et al., 2024), in `IWoL` it serves primarily as a representation learner; if desired, the same backbone can also be used to generate explicit messages at test time.

**Contribution:** *(i)* Firstly, this work proposes a novel representation learning framework for multi-agent coordination, *interaction-world latent* (`IWoL`) that encodes both inter-agent relations and privileged world information at a task level. *(ii)* Secondly, `IWoL` extends the line of communication MARL by introducing a communication scheduler based on graph-attention, which serves purely as a representation learner under an implicit communication scenario. In addition, this architecture naturally admits two variants: an *implicit* mode, where no messages are required at test time, and an *explicit* mode, where the learned messages are fed into the policy network for message-rich coordination. *(iii)* Lastly, to demonstrate the efficiency of `IWoL`, we compare the proposed solution with MARL baselines across four challenging multiple robotics testbeds, *including* autonomous driving (Li et al., 2022b), swarm-robot coordination (Pickem et al., 2017), bimanual dexterous hand manipulation (Chen et al., 2022), and multiple quadruped robots coordination (Xiong et al., 2024).

## 2 RELATED WORKS

**Cooperative Multi-agent Reinforcement Learning.** Transcending the single-agent environment, MARL has received attention to tackle robotics and other domains of the real world; the fact that each agent relies on a noisy and local observation of the environment further compounds coordination difficulties (Yang et al., 2023; He et al., 2022). To surmount these challenges and foster cooperative behavior, several approaches have been proposed that enable agents to learn effective strategies through the joint optimization of a shared team objective in online settings. The centralized training and decentralized execution (CTDE) framework is widely used as a promising alternative (Zhang et al., 2021). This framework leverages global state information during training to alleviate partial observability while each agent learns a decentralized policy that relies solely on local observations during execution (Yu et al., 2022a; Lowe et al., 2017; Qu et al., 2020; Rashid et al., 2020; Dou et al., 2022; Li et al., 2022a; Foerster et al., 2018; Li et al., 2021a; Shao et al., 2023; Wen et al., 2022; Zhu et al., 2024). This work introduces a novel MARL algorithm that models the interaction-world latent representation using a communication protocol, enabling robust performance in cooperative MARL tasks.

**Communication for Multi-agent Coordination.** Inter-agent communication is a linchpin to facilitate effective multi-agent coordination in MARL, fostering synergistic collaboration between agents. Most seminal works in MARL have centered on explicit communication protocols, where agents exchange messages to gain a clear understanding of others' context; for example, predefined full communication (Foerster et al., 2016; Sukhbaatar et al., 2016), partial communication (Wang et al., 2020; Yuan et al., 2022), learnable communication with adaptive gating mechanisms (Jiang & Lu, 2018; Singh et al., 2019; Ding et al., 2020; Hu et al., 2024b). These solutions frequently impose heavy communication burdens, making them unsuitable for real-world scenarios with limited bandwidth while also being vulnerable to adversarial attacks (Dafoe et al., 2020; Grimbly et al., 2021; Yuan et al., 2023). To address these challenges, we propose a communication protocol that learns inter-agent relations as an interactive latent. This design embeds coordination cues directly into the latent, enabling message-free deployment and decentralized, efficient inference.

**Representation for Multi-agent Reinforcement Learning.** Representation is critical for RL improvement, empowering agents to distill complex observations into useful abstractions that drive more informed decision-making. In MARL, individual agents use representation not only to better encode their local observations but also to facilitate the encoding of intricate inter-agent interactions or underlying world dynamics. Several works build representation in MARL, such as information reconstruction (Kim et al., 2023; Kang et al., 2025), auxiliary task-specific predictions (Shang et al., 2021), marginal utility function (Gu et al., 2022), self-predictive learning (Huh & Mohapatra, 2024; Feng et al., 2023), contrastive learning (Hu et al., 2024c; Song et al., 2023), and communication protocol (Jiang & Lu, 2018; Niu et al., 2021; Hu et al., 2024b). While these studies have advanced MARL representation learning, they focus narrowly on specific aspects, *e.g.*, local observation em-

bedding, inter-agent relationships, or world dynamics, without integrating them. Such approaches can lead to fragmented representations that may hinder agents from forming an understanding of their environment. This work proposes the `IWoL` framework that leverages privileged information during the training communication module to capture inter-agent relationships and world information.

# 3 BACKGROUNDS AND PROBLEM FORMULATION

**Decentralized Partially Observable MDP.** We formulate the MARL problem using a decentralized partially observable Markov decision process (Dec-POMDP) (Bernstein et al., 2002) $\mathcal{M}$, which is formally characterized by the tuple $\langle \mathcal{I}, \mathcal{S}, \mathcal{O}_i, \mathcal{A}_i, \mathcal{T}, \Omega_i, r_i, \gamma \rangle$, where $\mathcal{I} = 1, 2, \cdots, I$ denotes a set of agents. Here, $\mathcal{S}$ represents the global state space; $\mathcal{O}_i$ and $\mathcal{A}_i$ correspond to the observation and action spaces specific to agent $i$, respectively. The state transition dynamics are captured by $\mathcal{T} : \mathcal{S} \times \mathcal{A}_1 \times \cdots \times \mathcal{A}_I \mapsto \mathcal{S}$, while $\Omega_i : \mathcal{S} \mapsto \mathcal{O}_i$ specifies the observation function for agent $i$. Each agent $i$ receives rewards according to its reward function $r_i : \mathcal{S} \times \mathcal{A}_1 \times \cdots \times \mathcal{A}_I \mapsto \mathbb{R}$, and aims to maximize its discounted cumulative reward given by $R_i^t = \sum_{k=t}^{T-1} \gamma^{k-t} r_i^k$, with the temporal discount factor $\gamma \in [0, 1)$. Our framework is developed based on Dec-POMDP augmented with a communication protocol $P$. Consistent with previous literature in communication-based RL, we assume a deterministic MDP throughout this work, unless explicitly stated otherwise.

**Problem Setup.** We posit each agent $i$ has access only to its own local observation $o_i^t$ and must act without direct knowledge of the global state. Under such partial observability and interdependence, successful team performance hinges on effective coordination via synchronous communication. Concretely, we introduce a learnable protocol $P : M_1^0 \times \cdots \times M_I^0 \longmapsto M_1 \times \cdots \times M_I$ that transforms each agent's raw message $m_i^{t,0} \in M_i^0$ into a processed message $m_i^t \in M_i$. At every timestep $t$, all agents emit $m_i^{t,0}$, receive their corresponding $m_i^t$ from $P$, and then select actions.

Within the CTDE framework, our objective is to jointly learn the policies $\{\pi_i\}_{i \in \mathcal{I}}$ and the communication protocol $P$ so as to maximize the team's cumulative discounted return. To achieve it, we consider both implicit and explicit communication protocols.

**On-policy Optimization.** Our proposed solution is based on the on-policy optimization method, the multi-agent proximal policy optimization (MAPPO) algorithm (Schulman et al., 2017; Yu et al., 2022a). Based on this, we train all modules of a framework in an end-to-end manner under online settings. The loss for policy and value function is defined as follows:

$$\mathcal{L}_\pi(\phi_i) = -\min(p_i^t A_i^t, \text{clip}(p_i^t, 1 - \epsilon, 1 + \epsilon) A_i^t) - \mathcal{H}(\pi_{\phi_i}(a_i^t | o_i^t)), \tag{1}$$

where $\phi_i$ is parameter of agent $i$-th actor network, $p_i^t = \frac{\pi_{\phi_i}(a_i^t | o_i^t)}{\pi_{\phi_i^{\text{old}}}(a_i^t | o_i^t)}$ is the probability ratio, $A_i^t$ is an advantage estimate, and $\epsilon$ is a clipping parameter typically set to a small value. The additional entropy term $\mathcal{H}(\pi_{\phi_i}(a_i^t | o_i^t))$ encourages exploration by penalizing overly confident policies.

Next, the centralized critic parameters $\theta$ are updated via a clipped value loss to stabilize learning with the shared observation $\mathbf{o}^t = [o_1^t, \cdots, o_i^t]$ or state $s^t$, as follows:

$$\mathcal{L}_V(\theta) = \max\left(\ell_\delta\left(R^t - V_\theta(\mathbf{o}^t)\right), \ell_\delta\left(R^t - V_\theta^{\text{clip}}(\mathbf{o}^t)\right)\right), \tag{2}$$

$$\text{where } V_\theta^{\text{clip}}(\mathbf{o}^t) = V_{\theta_i^{\text{old}}}(\mathbf{o}^t) + \text{clamp}(V_{\theta_i}(\mathbf{o}^t) - V_{\theta_i^{\text{old}}}(\mathbf{o}^t), -\epsilon, \epsilon)$$

and $\ell_\delta(\cdot)$ is the Huber loss (Huber, 1992), which is less sensitive to outliers than a standard MSE loss.

# 4 INTERACTION-WORLD LATENT (IWoL)

In this section, we first show a motivating example of when explicit communication fails in 4.1, introduce `IWoL` in Section 4.2, and then explain how to train it in Section 4.3.

## 4.1 A MOTIVATING EXAMPLE

Figure 1 shows performance degradation in a simple traffic junction (Sukhbaatar et al., 2016) where multiple agents rely on explicit message passing to coordinate their movements. While unconstrained communication achieves

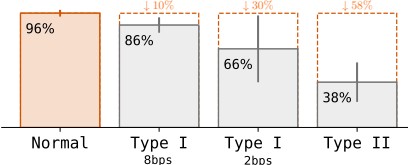

Figure 1: **A motivating example.** Explicit communication's performance degradation in two challenging scenarios: Type **I**. Bandwidth (bits per second) constraint; Type **II**. Communication attack.

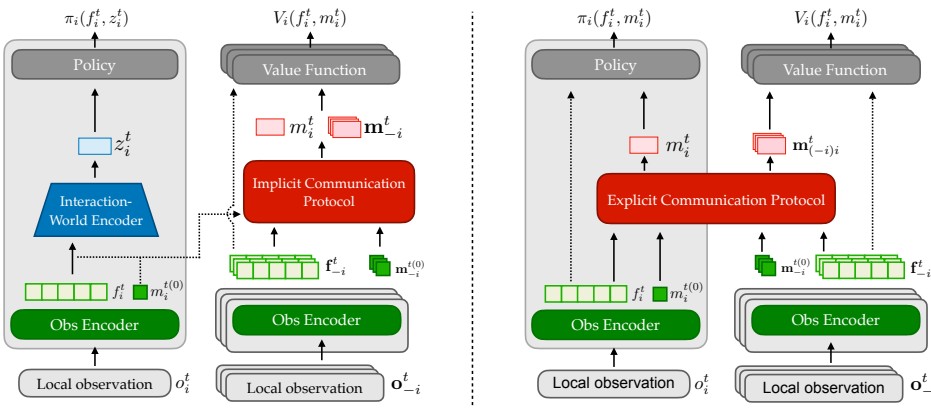

Figure 2: **Overview diagram of IWoL framework.** Grey box represents each agent. (*Left*) Implicit variation of IWoL. In this variation, each agent does not use communication messages at execution time. (*Right*) Explicit variation of IWoL. Policy directly uses a communication message from an explicit communication protocol. Note that IWoL's value function is decentralized, and it uses its own message and local embedding $V_{\theta_i}(m_i^t, f_i^t)$. Herein, $\cdot_{-i}$ means all agent's elements except $i$, and $\cdot_{(-i)i}$ includes all agent's elements including $i$.

nearly perfect success (Normal), imposing a bandwidth limit (Li & Zhang, 2024) reduces performance by about $10 \sim 30\%$ (Type I), and a message-corruption attack (Sun et al., 2023) leads to a severe performance drop (Type II), *i.e.*, approximately $60\%$. This dramatic degradation under realistic constraints and adversarial conditions reveals the fragility of explicit channels. To ensure robust multi-agent coordination, we thus advocate for the necessity of *implicit communication* in which message changes only happen within training. That is because the implicit communication method naturally overcomes these issues. Please see the Appendix D for details of this toy example.

## 4.2 INTERACTION-WORLD LATENT FOR MULTI-AGENT COORDINATION

**Our desiderata** are twofold: first, to avoid explicit message passing, but maintain team coordination under multi-agent dynamics and partial observability; and second, to maintain simplicity by refraining from additional modules, thereby enabling faster decision-making. Therefore, we learn a latent representation $z_i^t$ from local observations $o_i^t$, encoding inter-agent relations and task-specific world information. For this, we redesign the communication protocol to serve our learning objective.

**Architectural Design.** Figure 2 shows an overview for IWoL. First, each agent $i$ employs an observation encoder depicted in Figure 3 to transform its raw local observation $o_i^t$ into other modules. Specifically, $o_i^t$ is processed by a self-attention layer, which produces an intermediate embedding $f_i^t$. This embedding is then forwarded through an MLP, which outputs the initial message $m_i^{t(0)} \in \mathbb{R}^d$, denoted as round 0. It serves as the raw input to the subsequent communication protocol. Formally, we may write

$$\left[f_i^t, m_i^{t(0)}\right] = \text{Encoder}\left(\text{Attn}(o_i^t)\right)$$

where $\text{Attn}$ denotes the self-attention and $\text{Encoder}$ denotes the MLP.

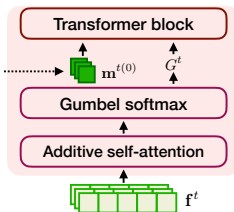

Figure 3: **Design for observation encoder.**

Next, the communication block (colored red in Figure 2 and detailed in Figure 4) is implemented with attention mechanisms to select neighbors and refine their messages in one go adaptively. Each agent's feature $f_i^t$ is fed into an additive-attention and $\text{GumbelSoftmax}$ block to produce a discrete adjacency mask, as communication graph $G^t$. This is a relationship graph that guides a Transformer block that performs $L$ rounds of attention-based message aggregation and refinement, mapping the initial message $m_i^{t(0)}$ directly to the final message $m_i^t$. This design allows the communication block to serve directly as a protocol for inter-agent coordination.

Figure 4: **Design for communication protocol.**

Lastly, the previously produced vectors, *e.g.*, communication message $m_i^t$ and intermediate embedding $f_i^t$, are used as input to policy and value function networks. We design a policy network $\phi_i$ and the value function network $\theta_i$ as a feed-forward layer. Herein, a policy network is set in a stochastic form.

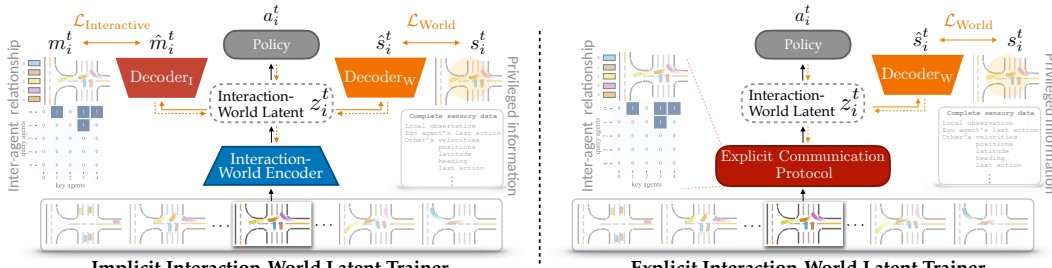

**Figure 5: Diagram for interaction-world Latent Modeling.** Solid and dotted lines denote forward and backward processes. The orange line implies only work in training. (*Left*) Im-IWoL uses the world and interactive decoder to reconstruct the privileged state and communication message $\hat{m}_i^t = \text{Decoder}_\text{I}(z_i^t)$. (*Right*) Ex-IWoL sets the latent to the message vector $z_i^t = m_i^t$, and then uses the world decoder to reconstruct the privileged state $\hat{s}_i^t = \text{Decoder}_\text{W}(z_i^t)$, thereby encouraging $z_i^t$ to embed the global information $s_i^t$.

**Graph-attention Communication Protocol.** Our communication protocol constructs the communication graph at each timestep in two stages. First, we obtain a learned adjacency graph $G_c^t$ using additive attention with Gumbel-Softmax as follows: $\{g_{c,ij}^t\}_{i,j=1}^I = \text{GumbelSoftmax}\left(\text{AddAtt}(\{f_k^t\}_{k=1}^I)\right)$ where $\text{AddAtt}(\cdot) = (a_{gg})^\top [W_{gg}f_i \| W_{gg}f_j]$, $a_{gg}$ is attention coefficient, and $W_{gg}$ is a learned weight matrix, respectively. Second, we build a physical mask $G_p^t$ to enforces spatial communication constraints for practicality $g_{p,ij}^t = \mathbb{1}(\|x_i^t - x_j^t\| \leq d_\text{comm})$, where $\mathbb{1}(\cdot)$ is an indicator function, $\mathbf{x}_i$ represents the position of agent $i$, and $d_\text{comm}$ denotes a maximum allowed communication distance. The final communication graph is then obtained as an element-wise product $G^t = G_c^t \odot G_p^t$, ensuring that only edges that are semantically relevant and physically feasible remain active.

Next, a Transformer block performs $L$ rounds of attention-based message propagation over $G^t$. Each round refines the intermediate messages $m_i^{t(l)}$ by aggregating information from their neighbors. After $L$ iterations, the final message $m_i^t = m_i^{t(L-1)}$ represents the agent's interaction information, encoding relational cues among agents.

**Interaction-world Latent Modeling.** Figure 5 visualizes an overview diagram about how to model IWoL explicitly and implicitly. Our goal is to build an interaction-world latent $z_i^t$ that captures inter-agent relationships and privileged world information for efficient multi-agent coordination. For this, the implicit IWoL (Im-IWoL) variant includes the following three modules.

$$\underbrace{z_i^t = \text{Encoder}_\text{IW}(f_i^t)}_{\text{Interaction}-\text{World Encoder}} \qquad \underbrace{\hat{m}_i^t = \text{Decoder}_\text{I}(z_i^t)}_{\text{Interaction Decoder}} \qquad \underbrace{\hat{s}_i^t = \text{Decoder}_\text{W}(z_i^t)}_{\text{World Decoder}}$$

The world decoder reconstructs the agent's privileged state $s_i^t$,[1] encouraging $z_i^t$ to embed task-specific global signals beyond local observation, whereas the interactive decoder reconstructs communication messages $m_i^t$, forcing $z_i^t$ to preserve inter-agent dependencies.

For the implementation of the explicit IWoL (Ex-IWoL), we eliminate the interaction-world encoder $\text{Encoder}_\text{IW}$ and instead reuse the explicit message emitted by the communication module itself, $z_i^t = m_i^t$, so we only use the world decoder $\text{Decoder}_\text{W}$ for training. While this design simplifies the latent construction and leverages explicit communication, Im-IWoL instead learns a representation from raw observations, which can easily capture more integrated interaction and world features.

> **Remark: Compatibility of IWoL's latent representation with two communication variants**
>
> In basic, IWoL is implemented as an *implicit* mode that only routes $m_i^t$ to the value network, not the policy network. *Explicit* communication mode routes $m_i^t$ to policy and value networks. The policy has direct access to relational signals between agents in the execution phase, while the value network uses the same structure to assign more accurate credit during training.

---

[1]The privileged state refers to task-specific information available only during training, such as proprioceptive information, other agents' complete internal states, and underlying physical parameters. Please see Appendix E.

(a) Intersection  (b) Parking lot  (c) Open Door Outward  (d) Open Bottle Cap  (e) Two Hand Underarm

(f) Navigation  (g) Material Transport  (h) Go1Gate  (i) Go1Sheep-Hard  (j) Go1Seesaw

Figure 6: **Visualization of the experimental scenarios where we evaluate `IWoL`.** The top row contains MetaDrive (a-b) and Bi-DexHands environments (c-e), and the bottom row includes Robotarium (f-g) and multiagent-quadruped-environments (h-j).

## 4.3 TRAINING

**Objective Function for Policy Training of `IWoL`.** To extract a policy, both variants of `IWoL` can be optimized in an end-to-end manner with a composite objective function as follows:

$$\underbrace{\mathcal{L}_\pi^{\text{Im}}(\phi_i) = \mathcal{L}_\pi^{\text{RL}}(\phi_i) + \lambda_\text{W}\mathcal{L}_\text{W} + \lambda_\text{I}\mathcal{L}_\text{I}}_{\text{Implicit Variants Objective}} \quad \text{and} \quad \underbrace{\mathcal{L}_\pi^{\text{Ex}}(\phi_i) = \mathcal{L}_\pi^{\text{RL}}(\phi_i) + \lambda_\text{W}\mathcal{L}_\text{W}}_{\text{Explicit Variants Objective}}, \tag{3}$$

where individual terms reflect distinct training signals.

RL policy objective $\mathcal{L}_\pi^{\text{RL}}(\phi_i)$ simply follows equation 1. World reconstruction objective $\mathcal{L}_\text{W}$ encourages latent $z_i^t$ (or $m_i^t$ in the explicit variant) to capture the privileged state $s_i^t$ that is only available during training; and communication message reconstruction objective $\mathcal{L}_\text{I}$ asks the interactive decoder to reproduce the original communication message.

$$\mathcal{L}_\text{W} = \left\|\text{Decoder}_\text{W}(z_i^t) - s_i^t\right\|_2^2 \quad \text{and} \quad \mathcal{L}_\text{I} = \left\|\text{Decoder}_\text{I}(z_i^t) - m_i^t\right\|_2^2$$

The coefficients $\lambda_s$ and $\lambda_m$ balance auxiliary supervision against the RL signal, then we set $\lambda_s > 0$ in both modes but choose $\lambda_m = 0$ for the explicit variant. We provide a pseudocode for the training algorithm and training details in the Appendix C.

## 5 EXPERIMENTS

In the following subsection, we introduce a suite of experiments designed to rigorously validate the efficacy of `IWoL` as a MARL framework. Specifically, our experiments will employ four multi-agent robotic tasks, selected to elucidate the five research questions. For our experiments, although we use shared or privileged information in training across all algorithms, this experimental setup is not fully observable since the input of the policy is directly limited. Please see the Appendix E and F for the experimental setup, detailed description of tasks, and additional empirical results, *including* ablations.

## 5.1 ENVIRONMENTAL SETUPS

We evaluate the proposed solution, `Ex-IWoL` and `Im-IWoL`, across four MARL environments: MetaDrive (Li et al., 2022b), Robotarium (Pickem et al., 2017), Bi-DexHands (Chen et al., 2022), and multiagent-quadruped-environments (MQE) (Xiong et al., 2024), visualized in Figure 6.

**MetaDrive** is a lightweight, large-scale simulator that features realistic traffic scenarios such as intersections and parking lots where multiple autonomous vehicles must coordinate to avoid collisions and reach their destinations. Since all ego agents have limited observability, this environment provides a strong benchmark for testing how well MARL methods enable communication for multi-agent coordination in dense traffic systems. We train each model under 1M timesteps.

**Robotarium** is a remotely accessible multi-robot testbed, used to evaluate physical coordination in real-world swarm robotics settings. We consider two tasks: simple navigation and material transport,

Table 1: **Performance evaluation**. We present a performance comparison across 10 tasks with four environments. These results are averaged over 4 seeds, and we report the two standard deviations after the $\pm$ sign. We highlight the best performance in **bold** and the second best in underlined . Note that if no $I$ is specified, $I = 2$. Basically, all tasks within the same testbed are ordered according to difficulty.

| Scenarios | | Metrics | MARL Baselines | | | | Proposed | |
|---|---|---|---|---|---|---|---|---|
| | | | MAPPO | MAT | MAGIC | CommFormer | Ex-IWoL | Im-IWoL |
| **MetaDrive** | **Intersection** ($I = 8$) | Rewards | 454.8 ±70.2 | 500.3 ±279.4 | 518.3 ±77.4 | 399.0 ±21.6 | **660.3** ±33.2 | 650.1 ±35.8 |
| | | Success (%) | 97.7 ±2.3 | 84.2 ±29.5 | 96.3 ±2.3 | 12.4 ±10.3 | **98.3** ±3.8 | 97.1 ±3.0 |
| | | Safety (%) | 94.8 ±5.2 | 76.0 ±35.4 | **99.1** ±0.9 | 9.6 ±4.8 | 98.3 ±3.8 | 97.1 ±3.0 |
| | **Parking lot** ($I = 5$) | Rewards | 327.4 ±211.7 | 605.8 ±163.4 | 371.2 ±14.3 | 527.6 ±195.6 | 619.7 ±79.6 | **808.6** ±51.0 |
| | | Success (%) | 30.3 ±15.9 | 55.3 ±15.3 | 26.2 ±3.7 | 43.9 ±12.5 | 54.1 ±8.8 | **63.7** ±9.8 |
| | | Safety (%) | 33.8 ±11.3 | 54.8 ±15.0 | 29.8 ±2.2 | 40.0 ±14.6 | 53.7 ±8.8 | **63.6** ±9.8 |
| | **Average rewards** | | 391.1 | 553.1 | 444.8 | 463.3 | 640.0 | **729.4** |
| **Robotarium** | **Navigation** ($I = 4$) | Rewards | −4.1 ±0.3 | −4.2 ±0.3 | −4.2 ±0.4 | −4.0 ±0.2 | −3.7 ±0.9 | **−3.5** ±0.4 |
| | | Safety (%) | 100.0 ±0.0 | 100.0 ±0.0 | 100.0 ±0.0 | 100.0 ±0.0 | 100.0 ±0.0 | 100.0 ±0.0 |
| | **Material transport** ($I = 4$) | Rewards | 2.7 ±0.8 | 3.11 ±1.2 | 2.6 ±0.2 | 1.2 ±2.2 | 3.6 ±0.1 | **3.8** ±0.1 |
| | | Safety (%) | 96.1 ±3.9 | 100.0 ±0.00 | 99.5 ±0.05 | 100.0 ±0.00 | 100.0 ±0.0 | 100.0 ±0.0 |
| | | Left materials | 18.3 ±7.9 | 12.4 ±10.8 | 28.6 ±18.4 | 27.4 ±11.0 | **4.7** ±6.0 | 5.0 ±5.0 |
| | **Average rewards** | | −0.7 | −0.5 | −0.8 | −1.4 | −0.05 | **0.15** |
| **Multi Quadrupeds** | **Go1Gate** | Rewards | 191.6 ±385.2 | 270.8 ±544.5 | 610.6 ±815.7 | −5.4 ±7.9 | **1570.2** ±247.8 | 1390.4 ±244.6 |
| | | Success (%) | 21.7 ±34.9 | 23.5 ±43.0 | 57.2 ±49.9 | 0.4 ±0.8 | **99.3** ±1.4 | 96.4 ±3.6 |
| | **Go1Sheep Hard** | Rewards | 1357.5 ±141.0 | 2834.9 ±1552.8 | 1284.9 ±10.2 | 1233.2 ±55.1 | 3340.9 ±506.3 | **6043.1** ±76.9 |
| | | Success (%) | 0.0 ±0.0 | 16.8 ±0.0 | 0.0 ±0.0 | 0.0 ±0.0 | 23.1 ±13.5 | **48.2** ±2.4 |
| | **Go1Seesaw** | Rewards | 4.3 ±13.2 | 56.0 ±24.2 | 79.7 ±52.7 | −37.6 ±5.1 | 84.1 ±51.0 | **114.1** ±41.4 |
| | | Success (%) | 0.0 ±0.0 | 1.7 ±2.8 | 3.6 ±7.0 | 0.0 ±0.0 | 6.8 ±5.9 | **15.3** ±7.8 |
| | **Average rewards** | | 517.8 | 1053.9 | 658.4 | 437.3 | 1665.1 | **2515.9** |
| **Bi-DexHands** | **Door Open Outward** | Rewards | 606.0 ±23.3 | 18.8 ±27.2 | 617.2 ±14.7 | 447.6 ±76.8 | 620.1 ±3.8 | **623.5** ±7.5 |
| | | Success (%) | 57.5 ±42.7 | 15.0 ±17.3 | 85.0 ±8.2 | 65.0 ±20.8 | 92.5 ±5.0 | **95.0** ±5.8 |
| | **Open Bottle Cap** | Rewards | 385.8 ±48.8 | 164.6 ±61.9 | 383.1 ±111.7 | 401.7 ±103.1 | 471.9 ±62.7 | **502.9** ±47.8 |
| | | Success (%) | 17.5 ±20.6 | 7.5 ±9.6 | 42.5 ±12.9 | 50.0 ±12.1 | 62.5 ±17.1 | **70.0** ±11.5 |
| | **Two Catch Underarm** | Rewards | 2.4 ±1.0 | 9.7 ±5.2 | 19.6 ±4.7 | 16.3 ±10.1 | 31.6 ±3.0 | **33.9** ±5.1 |
| | | Success (%) | 0.0 ±0.0 | 0.0 ±0.0 | 5.0 ±5.8 | 2.5 ±5.0 | 12.5 ±5.0 | **20.0** ±12.9 |
| | **Average rewards** | | 331.4 | 64.4 | 340.0 | 285.5 | 374.5 | 386.8 |

each requiring multiple robots to share local information and synchronize movement to avoid obstacles and reach common goals. Specifically, this environment tests cooperation and coordination ability through only inter-agent communication since ego agents do not use local perception devices (*e.g.*, camera or LiDAR). We set $0.5$M training timesteps.

**MQE** is based on Isaac Gym (Makoviychuk et al., 2021) that supports multi-agent tasks for quadrupedal robots. It includes cooperative tasks (*e.g.*, Narrow Gate, Seesaw, and Shepherding Sheep) where Unitree Go1 robots must coordinate to manipulate objects or navigate shared terrain. Agents operate under a hierarchical policy framework where high-level commands are issued over pre-trained low-level locomotion policies, allowing researchers to isolate the effect of coordinated planning and locomotion. For this task, evaluation is performed based on goal completion and auxiliary criteria such as safety and efficiency. We set $10$M training timesteps.

**Bi-DexHands** is a heterogeneous robotic manipulation and cooperation simulation built on Isaac Gym. Specifically, it is a multi-agent dexterous manipulation testbed, featuring two robotic Shadow Hands, each with $24$ degrees of freedom (DoF), enabling precise bimanual coordination. Tasks, two catch underarms, open door outward, and open bottle cap, require communication and contact-rich interactions between the two hands. We set the training period as $10$M, and its success rate is measured by a $10\%$ unit (*i.e.*, $0\%, 10\%, \cdots, 90\%, 100\%$).

**Baselines**. To comprehensively evaluate the performance of IWoL, we benchmark it against four established on-policy MARL algorithms. For on-policy, model-free methods, we include **MAPPO** (Yu et al., 2022a), which employs a centralized critic to address non-stationarity by leveraging global state information during training, while utilizing decentralized actors for execution. Furthermore, we consider **MAT** (Wen et al., 2022), which frames MARL as a sequence modeling problem, employing Transformer networks for both actors and critics. For communication-based MARL approaches, we assess **MAGIC** (Niu et al., 2021), which integrates a scheduler composed of a graph-attention encoder and a differentiable hard attention mechanism to dynamically determine communication timing and targets, alongside a message processor utilizing GATs (Veličković et al., 2018) to handle inter-agent messages efficiently. Furthermore, we evaluate the **CommFormer** (Hu et al., 2024b),

which conceptualizes the inter-agent communication architecture as a learnable graph, enabling adaptive and efficient information exchange among agents.

In our experiments, we employ four random seeds and represent $95\%$ confidence intervals with shaded regions in figures or standard deviations in tables, unless otherwise stated. All evaluations are performed under decentralized, partially observable, and goal-conditioned conditions, offering a comprehensive benchmark to evaluate how well the MARL algorithm enables scalable and adaptive coordination in diverse scenarios with different physical and strategic complexities.

## 5.2 EXPERIMENTAL RESULTS AND RESEARCH Q&A

**RQ1. How good is `IWoL` for MARL, compared to previous methods?**

**A:** `IWoL` variations achieve the `best` or `second-best` performance and success rate on most tasks.

Table 1 summarizes the aggregated experimental results for the 10 MARL benchmarks. We find that `IWoL` empirically outperforms prior MARL baselines. Surprisingly, `Im-IWoL` achieves the top score in 7 out of 10 tasks, with `Ex-IWoL` taking second in those; conversely, `Ex-IWoL` tops the remaining 3 tasks, with `Im-IWoL` finishing second, that is, together the two `IWoL` variants occupy the winner and runner-up in every task. We conjecture that the superiority of `Im-IWoL` arises not from communication efficiency itself, but from the jointly learned world encoding enabled. On the other hand, `Ex-IWoL` cannot replicate due to its reliance on explicit messages rather than learned latent structure. On average, `Im-IWoL` outperforms the strongest baseline by $+176.3$ points on MetaDrive and improves the Robotarium average reward from $-0.5$ to $+0.15$. In particular, our approach achieves up to $48.2\%$ and $20.0\%$ in MQE and Bi-DexHands, where previous baselines record near-zero success in three tasks (Go1Sheep, Go1Seesaw, and Two Catch Underarm). These results demonstrate that `IWoL` not only excels in standard MARL benchmarks but also effectively handles robotic manipulation tasks where existing MARL baselines fail.

**RQ2. Can `IWoL` maintain coordination performance under incomplete observations?**

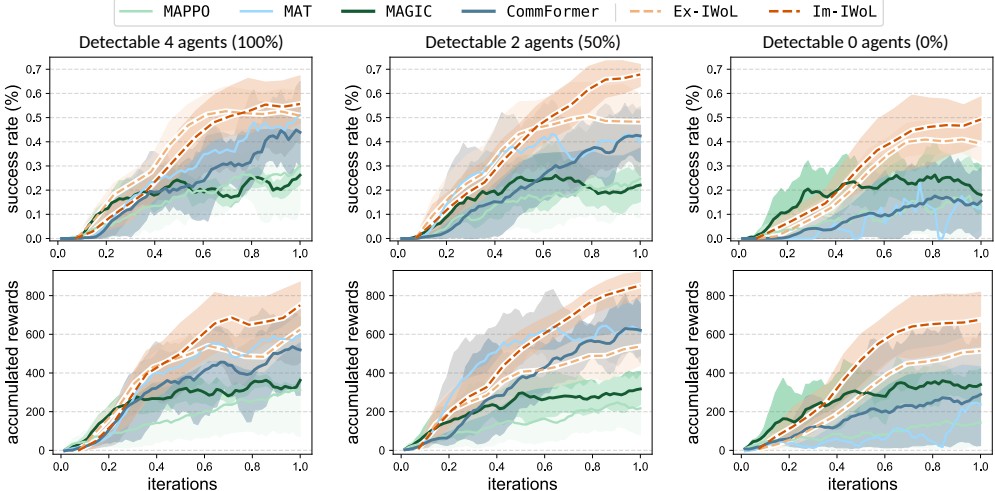

Figure 7: **Performance according to level of incomplete observations**. This presents the success rate and accumulated rewards as the number of detectable agents for the ego agent decreases in Parking Lot. This learning curve is plotted with the running average technique to differentiate baselines through a smoother line.

**A:** `IWoL` framework shows higher robustness than other baselines under incomplete observation scenarios.

In Figure 7, we present the success rate and accumulated reward as the number of detectable agents decreases. Such an experimental setup, as a communication-starved setting, is appropriate to study how well the proposed solution ensures robustness compared to baselines. This result demonstrates that both variants of `IWoL` maintain a clear margin over all baselines without severe drops in performance. Surprisingly, `Im-IWoL` always achieves above or about $50\%$ of success rate and small drops with 0 detectable agents, whereas `CommFormer` and `MAT` lose roughly twice as much. Additionally, `Ex-IWoL` leverages explicit message encoding, yielding slightly faster gains, whereas `Im-IWoL`'s implicit latent construction delivers superior asymptotic performance.

**RQ3. How effective is the world representation for the proposed and baseline solutions?**

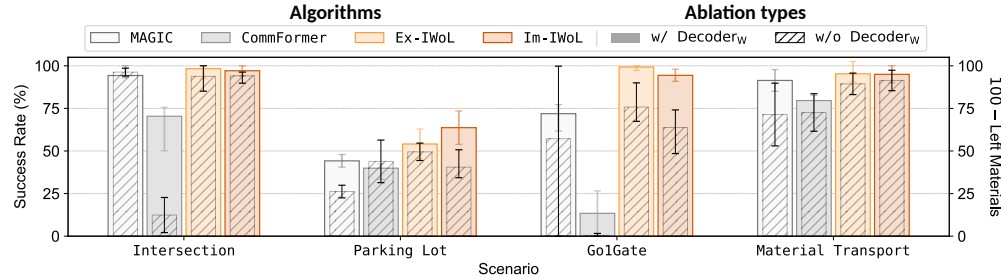

Figure 8: **Ablation study for world representation with Decoder$_W$.** The colored and dashed boxes indicate when there is no Decoder$_W$ for world representation and when there is , respectively; that is, the colored performance is original for `MAGIC` and `CommFormer`, and the dashed performance is original for `IWoL`. For the Material Transport task, we use inverted metrics (right $y$-axis) for readability.
**A:** World representation yields a noticeable performance gain for communication MARL solutions, and the Transformer-based communication module is competitive with other baselines on itself.

Figure 8 shows the ablation results of the world representation across communication-based MARL algorithms, that is, whether Decoder$_W$ is used during training. First, comparing IWoL variants with other baselines, we can see that it is competitive even without world representation embedding. This result implies that our Transformer-based communication protocol is powerful compared to other explicit communication baselines. Next, juxtaposing w/Decoder$_W$ and w/oDecoder$_W$ , we confirm that variants achieve better performances in most tasks and algorithms. Such empirical results directly demonstrate how effective and beneficial the world representation is for MARL training. Note that `IWoL` without its communication protocol and world latent is `MAPPO`.

**RQ4. Can `IWoL` maintain its performance as well in a large multi-agent system?**

Table 2: **Scalability test.** We provide a success rate according to the number of agents in Intersection.

| Algorithm | Success rate according to # of agents | | | | | # of agents | Baselines | |
|---|---|---|---|---|---|---|---|---|
| | 4 | 8 | 16 | 32 | 48 | | MAPPO | MAGIC |
| Im-IWoL | $99.5\%_{\pm 0.5}$ | $97.1\%_{\pm 3.0}$ | $95.2\%_{\pm 9.8}$ | $92.8\%_{\pm 16.5}$ | $86.6\%_{\pm 19.7}$ | $|\mathcal{I}| = 4$ | $98.5_{\pm 3.5}$ | $99.5_{\pm 0.5}$ |
| Ex-IWoL | $99.1\%_{\pm 1.0}$ | $98.3\%_{\pm 3.8}$ | $98.0\%_{\pm 11.2}$ | $95.5\%_{\pm 19.0}$ | $91.0\%_{\pm 28.1}$ | $|\mathcal{I}| = 48$ | $33.5_{\pm 36.5}$ | $75.0_{\pm 30.4}$ |

**A:** `IWoL` can maintain coordination performance decently as the agent population increases.

Table 2 reveals that coordination remains with good success rate with four agents, 99.5% for `Im-IWoL`, 99.1% for `Ex-IWoL`, and only marginally degrades as the team size doubles repeatedly, success stays above 95% up to 16 agents and above 92% (Im) / 95% (Ex) even at 32 agents. Remarkably, with 48 agents, `Ex-IWoL` still solves 91.0% of instances and `Im-IWoL` maintains 86.6%. In contrast, baselines suffer from a substantial performance drop from 4 to 48 agents. These observations confirm that `IWoL` has the potential to be adopted in large-scale MARL.

**RQ5. Is `Im-IWoL` superior to the previous implicit communication branches?**

Table 3: **Performance comparison with `ICP`.** We evaluate the performance of `ICP` across three variants.

| Scenarios | Metrics | ICP-Dec | ICP-Sum | ICP-Monotonic | Im-IWoL |
|---|---|---|---|---|---|
| **Parking Lot** | Rewards | $227.5_{\pm 30.6}$ | $289.5_{\pm 116.7}$ | $391.6_{\pm 236.2}$ | $\mathbf{808.6}_{\pm 51.0}$ |
| | Succ. Rate (%) | $15.7_{\pm 5.3}$ | $20.6_{\pm 8.4}$ | $35.6_{\pm 18.0}$ | $\mathbf{63.7}_{\pm 9.8}$ |
| **Go1Gate** | Rewards | $130.7_{\pm 91.0}$ | $783.3_{\pm 495.1}$ | $710.9_{\pm 538.2}$ | $\mathbf{1390.4}_{\pm 244.6}$ |
| | Succ. Rate (%) | $5.8_{\pm 6.6}$ | $70.5_{\pm 15.5}$ | $67.8_{\pm 22.0}$ | $\mathbf{96.4}_{\pm 3.6}$ |
| **Door Open Outward** | Rewards | $178.2_{\pm 30.1}$ | $495.5_{\pm 25.8}$ | $518.6_{\pm 31.0}$ | $\mathbf{623.5}_{\pm 7.5}$ |
| | Succ. Rate (%) | $7.5_{\pm 9.6}$ | $52.5_{\pm 12.6}$ | $45.0_{\pm 14.1}$ | $\mathbf{95.0}_{\pm 5.8}$ |

**A:** Although such a claim is not our focus, `Im-IWoL` outperforms the previous implicit method.

Table 3 demonstrates that our solution yields substantially higher rewards and success rates across all scenarios. Herein, `ICP` (Wang et al., 2025) works like inverse modeling, agents use predefined 'scouting' actions to encode messages and recover them by inverse mapping under ideal broadcast and decoding assumptions. In contrast, `Im-IWoL` learns the interaction-world representation using a communication module during training, fusing privileged state and inter-agent relations. At deployment, `Im-IWoL` does not need additional modules, *e.g.*, communication or inference model.

Lastly, we would like to clarify that `Im-IWoL` is fundamentally different from previous branches; in other words, this claim is not our focus and core in this work. See the Appendix B and F for an extended literature survey of previous branches and experimental details.

## 6 CONCLUSION

This work presented `IWoL`, a unified representation-learning and communication framework for cooperative MARL. This learns a compact latent that jointly captures inter-agent relations and privileged task information through a representation-oriented communication protocol. Extensive experiments show that `IWoL` variants attain best performance in 10 out of 10 robotic tasks.

**Closing Remarks:** One especially appealing property of `IWoL` is its **versatility**—allowing practitioners to toggle between message-free (implicit) and message-rich (explicit) coordination without additional modules. Given the notorious sensitivity of large-scale MARL, we believe that offering a drop-in solution that is both *efficient and simple* is a timely contribution to the community.

## FUTURE DIRECTIONS

Recently, generalizability has been a key challenge in the machine learning domain, and its promising workaround is representation learning, achieving notable success in real-world applications, *e.g.*, foundation and omni models (Black et al., 2024; Intelligence et al., 2025). For single-agent RL, diverse representation learning methods have been introduced, *e.g.*, successor features (Agarwal et al., 2024; Sikchi et al., 2024; Barreto et al., 2017), forward-backward representations (Touati & Ollivier, 2021; Touati et al., 2022), quasimetrics (Wang et al., 2023; Valieva & Banerjee, 2024), bisimulation (Zhang et al., 2020; Kemertas & Aumentado-Armstrong, 2021; Hansen-Estruch et al., 2022), temporal distance (Park et al., 2024; Bae et al., 2024; Lee & Kwon, 2025b), and contrastive objectives (Zheng et al., 2023; Myers et al., 2024), each contributing to more generalizable RL agents.

Unlike the single-agent setting, where representations can be extracted from a fully observable environment, extending this perspective to MARL is not a trivial problem. That is because MARL requires handling diverse information such as inter-agent relationships, their role structures, and shared world dynamics under a partially observable MDP. While some methods leverage one of these aspects in isolation, there has been limited progress in integrating them into a unified representation that captures the factors facilitating team coordination. Alternatively, to develop the generalizability, ad-hoc teamwork (Wang et al., 2024a; Gu et al., 2022) and zero-shot adaptation (Jha et al., 2025) have highlighted the importance of enabling agents to coordinate with previously unseen partners. They suggest that increasing the diversity of self-play scenarios can improve agents' adaptation and generalization ability.

While our approach is not designed as a direct solution to these challenges, it provides a unified latent representation that could support such functionalities. From this perspective, we envision several meaningful directions for future research:

- How can we leverage a representation learning framework from a single-agent to a multi-agent setting?

- Can we construct an omni-representation that jointly encodes diverse information in a scalable manner?

- What information and mechanisms facilitate rapid role alignment and policy adaptation in multi-agent dynamics?

In conclusion, this work is an initial step toward bridging representation learning and generalizable multi-agent coordination, instead of attempting a definitive solution to these open challenges. Our framework provides a foundation upon which future work can build more adaptive and versatile strategies using a unified latent representation. Finally, we hope this perspective stimulates further exploration into *omni-representation* and its role in enabling robust open-world multi-agent systems.

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

# Appendix

## CONTENTS

# A   Miscellaneous

## A.1   Summary of Notations

**Dec-POMDP elements**

| Notation | Description | Notation | Description |
|---|---|---|---|
| $\mathcal{I}$ | agents set | $i$ | agent index |
| $I$ | the number of agents | $\gamma \in [0,1)$ | discount factor |
| $\mathcal{S}$ | state space | $s^t$ | state |
| $\mathcal{O}_i$ | observation space of agent $i$ | $o_i^t$ | local observation of agent $i$ |
| $\mathcal{A}_i$ | action space of agent $i$ | $a_i^t$ | action of agent $i$ |
| $\mathcal{T}$ | state transition model | $\Omega_i$ | observation function of agent $i$ |
| $r_i$ | reward function for agent $i$ | $R_i^t$ | Return of agent $i$ |

**Algorithm elements**

| Notation | Description | Notation | Description |
|---|---|---|---|
| $M_i^0$ | space of initialized messages | $M_i$ | space of processed messages |
| $m_i^{t(0)}$ | initialized message from agent $i$ | $m_i^t$ | final processed message for agent $i$ |
| $m_i^{t(l)}$ | message for agent $i$ at round $l$ | $L$ | total communication rounds |
| $f_i^t$ | intermediate embedding of agent $i$ | $g_{ij}^t$ | relationship between agents $i$ and $j$ |
| $G^t$ | topology graph | $a_g g$ | weighting vector of the additive attention |
| $z_i^t$ | interaction-world latent for agent $i$ | $s_i^t$ | privileged state for agent $i$ |
| $\hat{m}_i^t$ | reproduced message of agent $i$ | $\hat{s}_i^t$ | reproduced privileged state of agent $i$ |

**RL Training**

| Notation | Description | Notation | Description |
|---|---|---|---|
| $\phi_i$ | policy parameters for agent $i$ | $\theta$ | value function parameters |
| $\lambda_{\mathrm{W}}$ | balancing coefficient for world latent loss | $\lambda_{\mathrm{I}}$ | balancing coefficient for interactive latent loss |
| $\epsilon$ | clip coefficient of PPO loss | $\delta$ | threshold of Huber loss |
| $K$ | the number of updates | $T$ | max steps of an episode |
| $B$ | batch size | $\mathcal{D}$ | online replay buffer |
| $\eta$ | learning rate | | |

## A.2   System Specification

| | |
|---|---|
| CPU | AMD EPYC 7713 64-Core |
| GPU | RTX A6000 & A5000 |
| Software | CUDA: 11.8, cudnn: 8.7.0, python: 3.8 |
| PyTorch | 2.1.0 (MetaDrive), 2.0.1 (Robotarium), 2.4.1 (MQE and Bi-hand Dexterous) |

## B  EXTENSIVE RELATED WORKS

### B.1  USAGE DIFFERENCES OF TRANSFORMER IN MAT, COMMFORMER, AND IWOL

MAT (Wen et al., 2022) first formulates cooperative MARL as a sequence modeling problem, employing a full encoder–decoder Transformer to map a sequence of agents' joint observations to a sequence of optimal actions. The encoder uses stacked self-attention and MLP blocks to capture high-level inter-agent dependencies, while the decoder generates each agent's action auto-regressively under a causal mask that restricts attention to preceding agents. This design yields linear complexity in the number of agents and comes with a monotonic performance improvement guarantee.

CommFormer (Hu et al., 2024b) basically builds on MAT by explicitly learning a sparse communication graph. It introduces a learnable adjacency matrix and incorporates this graph both as an explicit edge embedding in the attention score computation and as a hard mask to gate message passing. Consequently, its Transformer encoder and decoder are conditioned on both causal order and dynamic connectivity, enabling concurrent optimization of communication architecture and policy parameters in an end-to-end fashion.

In IWoL, the Transformer block is repurposed exclusively as the communication processor, instead of a policy or value function. After each agent's local observation is encoded, a Graph-Attention Transformer applies multi-head scaled dot-product attention over a communication graph to refine interactive embeddings.

We summarize the usage differences of the Transformer in these methods as follows.

- **MAT**: Transformer serves as the joint policy network, with encoder–decoder modeling and causal masking.

- **CommFormer**: Transformer integrated with a learnable, sparsity-controlled communication graph to gate attention.

- **IWoL**: First work to design the communication processor itself as a graph-attention Transformer, combining attention-based message encoding.

### B.2  IMPLICIT COMMUNICATION AS INVERSE MODELING

**Implicit communication channel.** A growing body of work has explored implicit coordination without explicit messaging. For example, (Li et al., 2023) trains agents to gradually shift from using explicit messages to purely tacit cooperation. Other approaches learn latent coordination through structured interactions, such as inferring dynamic coordination graphs, or via environment-mediated signaling (Li et al., 2021b). Notably, some methods endow agents with implicit communication abilities by leveraging shared environment cues (Wang et al., 2024b; Shaw et al., 2022; Grupen et al., 2022; Wang et al., 2025). However, these implicit communication frameworks typically lack a dedicated learned messaging architecture and often focus on narrow aspects of coordination, which is similar to inverse modeling through environmental to behavioral cues.

**Machine theory of mind.** Another line of research draws on cognitive reasoning (Baron, 1997; Premack & Woodruff, 1978), where agents explicitly model each other's beliefs, intentions, or roles. For example, theory-of-mind (ToM) approaches have been combined with social incentives like guilt aversion to encourage cooperation (Rabinowitz et al., 2018; Nguyen et al., 2020; Leroy et al., 2023). Such agents maintain internal beliefs about what others will do and even what others believe they will do, implementing recursive reasoning to adjust their policies (Wang et al., 2022; Oguntola et al., 2023; Doshi et al., 2020). Similarly, brain-inspired ToM models use structured networks with dedicated modules for perspective-taking, policy inference, and action prediction, that is, essentially mimicking human mentalizing by explicitly predicting others' observations and actions (Zhao et al., 2022).

**Agent modeling and action prediction.** A third branch of related work focuses on agents forecasting or simulating their counterparts' behavior to improve coordination (He et al., 2016; Yu et al., 2022b; Ganzfried & Sandholm, 2011). Recent methods explicitly model other agents' policies or future actions as part of the decision process. For example, (Gupta et al., 2023) introduces conformal prediction sets to model other agents' actions with high confidence, providing each agent with a set of likely moves of others before acting. Interactive MDPs explicitly construct recursive belief

models of other agents' hidden states and policies to guide planning, whereas interactive latent coding methods learn compact embeddings of observed interaction dynamics without forming full belief hierarchies (Xie et al., 2021; Hoang & Low, 2013). Other approaches give agents an explicit planning capability, such as an episodic future thinking mechanism where an agent infers each partner's latent character and then simulates future trajectories of all agents to select an optimal action (Lee & Kwon, 2024a;b; 2025a). Likewise, fact-based agent modeling trains a dedicated belief inference network that uses an agent's own observations and rewards to reconstruct the policies of other agents through a variational auto-encoder (Fang et al., 2023). These techniques incorporate additional structures to predict or encode other agents' states, essentially bolting on an extra layer of agent-specific reasoning.

`Im-IWoL` departs from all three lines by folding communication and modeling into a single Transformer block that is used only during centralized training. Messages circulate through the Transformer while the topology and latent code are being learned, but at test time, each agent discards the channel and relies solely on the cached latent embeddings produced by its local encoder; no decoder, simulator, or action-based signalling set is required. This makes IWoL, to our knowledge, the first implicit-communication framework that needs zero additional modules at deployment while still endowing agents with an internal world latent that unifies inter-agent relations and global task information. Consequently, `Im-IWoL` inherits the bandwidth-free, attack-resistant advantages of prior implicit schemes, yet retains the architectural simplicity and real-time footprint of a feed-forward network.

### B.3 PRIVILEGED INFORMATION IN LEARNING TO CONTROL

Privileged information, which is signals available at training but not at deployment, has been widely adopted for learning to control (Cai et al., 2024; Li et al., 2024b; Ndousse et al., 2021; Salter et al., 2021; Hu et al., 2024a; Kamienny et al., 2020; Li et al., 2024a; Huang et al., 2025), *e.g.*, MARL under partial observability, world model training, and policy improvement. For MARL, it has been introduced both as a training-only information for exploration or credit assignment (Cai et al., 2024; Li et al., 2024b; Ndousse et al., 2021), as well as an attention or distillation signal to align decentralized policies with global information (Salter et al., 2021; Li et al., 2024a). For world-model training, privileged sensing has guided the training of latent dynamics and representation alignment to improve sample efficiency and safety (Hu et al., 2024a; Huang et al., 2025). Furthermore, robustness techniques such as privileged information dropout prevent over-reliance on non-deployable signals and encourage policies to learn deployable features (Kamienny et al., 2020). Across these settings, the unifying principle is that privileged signals are leveraged as auxiliary supervision during training, but removed at deployment to ensure fairness and applicability. In our work, we follow this paradigm by using a privileged decoder to align shared latent representations, while keeping execution fully decentralized and based only on local observations.

## C  TRAINING DETAILS

### C.1  PSEUDOCODE FOR IWoL

---

**Algorithm 1** Training IWoL

---

1: **Require**: the number of episode $K$, the number of agents $I$, max steps of an episode $T$, batch size $B$, replay buffers $\mathcal{D}_i$ for each agent $i$, and learning rate $\eta$
2: **Initialize**: actor parameters $\Phi = \{\phi_1, \phi_2, \cdots, \phi_I\}$, and critic parameters $\Theta = \{\theta_1, \theta_2, \cdots, \theta_I\}$
3: Initialize buffer $\mathcal{D}_i$ and loss function $\mathcal{L}(\theta_i), \mathcal{L}(\phi_i)$
4: **for** episode $= 1, K$ **do**
5:     Reset privileged state $s$, observations $\mathbf{o}$
6:     `#Roll-out trajectories` --------------------------------------
7:     **for** $t = 1, T_{\max}$ **do**
8:         Local observation embedding: $\mathbf{f}^t, \mathbf{m}^{t,0} \leftarrow \text{Encoder}(\text{SelfAttn}(\mathbf{o}))$
9:         Scheduling communication: $G^t \leftarrow \text{GumbelSoftmax}(\text{AddAtt}(\mathbf{f}^t))$
10:       Message processing with $L$ hops: $\mathbf{m}^t \leftarrow \text{Transformer}(\mathbf{m}^{t,0}, G^t, L)$
11:       `# Building latent for coordination` ------------------------
        Im-IWoL: $\mathbf{z}^t \leftarrow \text{Interactive World Encoder}(\mathbf{f}^t)$       if Ex-IWoL: $\mathbf{z}^t \leftarrow \mathbf{m}^t$
12:       `# Policy and Value Function` --------------------------------
        Im-IWoL: $\mathbf{a}^t \leftarrow \Pi_\Phi(\mathbf{z}^t)$ and $\mathbf{v}^t \leftarrow \mathbf{V}_\Theta(\mathbf{m}^t, \mathbf{f}^t)$    if Ex-IWoL: $\mathbf{a}^t \leftarrow \Pi_\Phi(\mathbf{z}^t, \mathbf{f}^t)$
13:       Perform action $\mathbf{a}^t$, then transit state $s^{t+1}$, receive $\mathbf{r}^t$ and $\mathbf{o}^{t+1}$
14:       Store transition $(o_i^t, a_i^t, \pi_{\phi_i}(a_i^t|o_i^t), V_{\theta,i}(o_i^t), o_i^{t+1})$ in $\mathcal{D}_i$ for each agent $i$
15:       $t \leftarrow t + 1$
16:     **end for**
17:     `#Return estimation` ------------------------------------------
18:     **for** $i = 1, I$ **do**
19:         **for** $t = 1, T_{\max}$ **do**
20:            $R_i^t = r_i^t + \gamma R_i^{t+1}$ **if** $o_i^{t+1} \neq$ terminal from $\mathcal{D}_i$
21:         **end for**
22:       `#Decentralized network update` -----------------------------
23:       Reconstruct privileged state: $\hat{s}_i^t \leftarrow \text{Decoder}_W(z_i^t)$
24:       Calculate $\mathcal{L}_{\text{World}} \leftarrow \lambda_W \cdot \text{MSE}(s_i^t, \hat{s}_i^t)$
25:       Reconstruct communication message: $\hat{m}_i^t \leftarrow \text{Decoder}_I(z_i^t)$      # only Im-IWoL
26:       Calculate $\mathcal{L}_{\text{Interactive}} \leftarrow \lambda_I \cdot \text{MSE}(m_i^t, \hat{m}_i^t)$     if Ex-IWoL, $\mathcal{L}_{\text{Interactive}} = 0$
27:       Calculate $\mathcal{L}(\phi_i)$ and $\mathcal{L}(\theta_i)$ using $\mathcal{D}_i$ with equation 1 and equation 2
28:       $\mathcal{L}(\phi_i) \leftarrow \mathcal{L}(\phi_i) + \mathcal{L}_{\text{Interactive}} + \mathcal{L}_{\text{World}}$
29:       Update $\theta_i \leftarrow \theta_i - \eta \nabla \mathcal{L}(\theta_i)$ and $\theta_i \leftarrow \phi_i - \eta \nabla \mathcal{L}(\phi_i)$
30:     **end for**
31:     episode $\leftarrow$ episode $+ 1$
32: **end for**

---

### C.2  IMPLEMENTATION DETAILS

**Multi-threaded synchronous policy optimization.** Our training implementation employs $N_{\text{threads}}$ parallel worker threads, each running an independent environment and collecting fixed-length trajectory segments (Singh et al., 2019). After each segment, every worker computes policy and value function losses to derive gradients, which are then synchronously aggregated across all threads. We average these gradients and perform a single parameter update using the Adam optimizer (Kingma & Ba, 2014). To promote exploration, we include an entropy bonus with weight 0.01 and train the value head with a mean-squared error loss (Schulman et al., 2017). Gradients are clipped to a maximum norm of 0.5 to ensure stable updates (Koloskova et al., 2023). Updated network parameters are then broadcast back to all workers before the next rollout cycle.

**Decentralized value function for IWoL variants.** In our approach, we adopt privileged state information for each agent, enabling it to alleviate heavy dependence on a centralized critic in previous methods. Consequently, every agent learns its own value function in a fully decentralized fashion (*i.e.*, using only its local, augmented observations) thereby avoiding the communication

overhead and staleness issues inherent to a centralized value estimate, while still benefiting from the extra privileged information to maintain sample efficiency and training stability.

**Self-attention in the observation encoder.** Self-attention can capture dependencies among features within each agent's local observation. Therefore, it allows the later layer to identify salient objects and their relations, providing relational cues even before inter-agent communication occurs. Moreover, when historical observations are included as input, the self-attention mechanism effectively captures long-term temporal dependencies across time steps. We believe that this component can be replaced on the recurrent neural network variants, *e.g.*, LSTM or GRU.

**Additive-attention in communication module.** Additive-attention provdies a stable scoring mechanism to construct pairwise relevance between features, which is widely used for dynamic interaction graphs. Unlike dot-product attention, it is less sensitive to feature scaling and therefore produces smoother gradients when learning dynamic communication graphs. This property is particularly useful for multi-agent scenarios, where interaction patterns frequently change depending on the agents' spatial and behavioral context.

**Gumbel-Softmax in communication module.** `Gumbel-softmax` component enables differentiable discrete sampling of communication links. In other words, it allows the communication graph to be trained end-to-end by combining with RL training objectives. As a result, the model learns structured and adaptive message-passing topologies without the need for manually designed communication rules.

**Training Details.** Additionally, we employ two techniques for efficient updates: value normalization (Mehta et al., 2023) and active masking (Stolz et al., 2024). Value normalization keeps the target values $R_i^t$ on a consistent scale, improving stability and training convergence. Next, optionally employ active masks in the critic to ensure that irrelevant states or features do not excessively affect the value function estimates. This can help the critic focus on relevant state dimensions and reduce training variance.

**Fairness of Baseline Implementation.** To ensure a fair comparison, we modify some practical implementations of all baselines so as to `IWoL`:

- **Use of privileged/global information:** Every CTDE baseline (`MAPPO`, `MAT`, `MAGIC`, `CommFormer`) was given access to the exact same privileged signal, either the ground-truth global state $s^t$ or the concatenated shared observation $\mathbf{o}^t = [o_1^t, \ldots, o_I^t]$, when fitting its value function. In other words, *all* methods can use the same global information in a training phase, so that any performance gap cannot be attributed to unequal access to global information.

- **Local-observation encoding:** To match `IWoL`'s encoder capacity, we equipped each baseline with an identical local-observation embedding pipeline: each agent's raw observation $o_i^t$ is first projected into a fixed-dimensional feature, then passed through a self-attention layer or a lightweight RNN. We held all embedding hyperparameters constant across methods. This design guarantees that improvements stem from our communication–world-latent architecture rather than from extra encoding power.

- **Hyperparameter:** To ensure a fair comparison, we aligned both algorithmic and environment parameters across all methods. For algorithmic hyperparameters, we adopted the original settings reported in each paper (*i.e.*, learning rates, network architectures, `PPO`-dependent parameters such as clipping $\epsilon$, value-loss, and entropy weights). In addition, we fixed batch size, hidden dimensions, number of attention heads, and all PPO-specific coefficients to be identical for `IWoL` and every baseline. Next, for environmental hyperparameters, all training parameters, *e.g.*, maximum episode length, reward scaling, observation noise, and number of parallel environments, were standardized across experiments.

# D  TOY EXAMPLE DETAILS

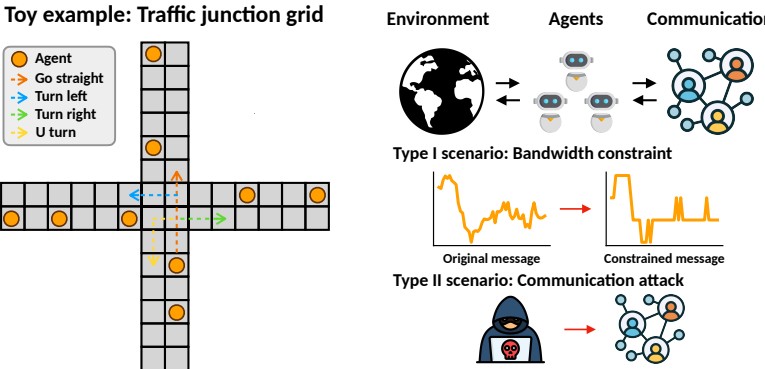

Figure 9: **A diagram for a toy example.** In this example, we adopt a simple traffic junction task and two challenging scenarios in communication MARL.

**Simple Traffic Junction.** We consider a 4-way traffic junction on a $14 \times 14$ grid. At each time step, new cars enter from each of the four entry points with probability $p_{\text{arrive}}$, up to a maximum of $N_{\text{max}} = 10$ concurrent cars. Each car occupies a single cell and is randomly assigned one of four routes (keeping to the right side): straight, left turn, right turn, or U turn. At each step, an agent chooses between two actions: moving one cell or staying in place.

Agents aim to exit the grid upon reaching their destination. They can get a one-hot encoding of its route ID ($\{n, l, r\}$) as an observation without others' information. Agents overcome their partial observability by explicitly communicating with all others.

The total team reward at time $t$ is as follows:

$$r(t) \;=\; C^t \, r_{\text{coll}} \;+\; \sum_{i=1}^{N^t} \tau_i \, r_{\text{time}},$$

where $C^t$ the number of collisions, $N^t$ the number of cars present, $r_{\text{coll}}$ is a collision penalty, $r_{\text{time}}$ is a a time penalty, and $\tau$ is the number of steps since that car entered. Episodes run for 40 steps and are marked as failures if any collision occurs.

**Type I: Bandwidth Constraint.** In *Type I*, we limit the number of bits used to transmit each continuous message vector $m_i^t$. Originally, we used 32 bits per element. Here, we quantize to only $n \in \{8, 2\}$ bits per element and measure the resulting performance drop. At execution time, we consider the following bandwidth constraints:

```
def bit_per_sec_const(message, n):
    levels = 2**n
    # scale to [0, levels-1], round, then rescale to [0,1]
    const_m = torch.round(message * (levels - 1)) / (levels - 1)
    return const_m
```

for $n = 8$ (Type I - 8bps) or $n = 2$ (Type I – 2bps).

**Type II: Communication Attack.** In *Type II*, an adversary intercepts and corrupts each agent's message before delivery. Concretely, for every transmitted $m_i^t$, we sample a random vector $\hat{m}_i^t \sim \mathcal{U}$ and deliver $\hat{m}_i^t$ in place of the true message. This simulates a worst-case message corruption scenario, and we report the performance degradation under this attack.

# E   EXPERIMENTAL DETAILS

This appendix section describes the task and details of a Markov decision process over 10 scenarios.

**Remark.** Each task requires extensive notation so that some symbols may overlap with those used in the main text. Readers are therefore advised to consult each MDP specification and notation in the context of its own environment.

## E.1   METADRIVE

(*Limited observability, high-dimensional observation space, large-scale, cooperative, and competitive*) MetaDrive is an autonomous driving simulator formulated as an MDP. Each vehicle simultaneously learns a driving policy, a value function, and a communication protocol.

### E.1.1   MARKOV DECISION PROCESS

**Observation.** In particular, at each timestep $t$, agent $i$ receives an observation $o_i^t$ composed of three parts: ego-vehicle information, surrounding vehicles' information, and navigation cues.

- Ego-vehicle information: $\left[\theta_i,\ \phi_i,\ v_i,\ d_i^{\text{left}},\ d_i^{\text{right}}\right]$ where $\theta_i$ is steering angle, $\phi_i$ is heading, $v_i$ is speed, and $d_i^{\text{left}}, d_i^{\text{right}}$ are distances to left/right lane boundaries
- Surrounding vehicles' information: relative positions and velocities of up to $N_{\text{obs}}$ nearest agents, sensed via a LiDAR with $N_{\text{laser}}$ beams covering a $d_{\text{laser}}$.
- Navigation cues: A vector encoding direction and distance to the destination.

**Actions.** Each agent selects a continuous action $a_i^t = [\,\alpha_i^t,\ \beta_i^t\,]$, where $\alpha_i^t \in [-1, 1]$ controls steering (left:$-1$, right:$+1$), and $\beta_i^t \in [-1, 1]$ controls acceleration (brake:$-1$, throttle:$+1$).

**Reward.** We design the individual reward into three components:

$$r_i^I = c^{\text{driving}}\, r_i^{\text{driving}}\ +\ c^{\text{speed}}\, r_i^{\text{speed}}\ +\ r_i^{\text{termination}},$$

where driving progress $r_i^{\text{driving}}$ is the forward distance traveled along the lane between $t$ and $t + 1$, agility reward $r_i^{\text{speed}} = \frac{v_i^t}{v_{\text{max}}}$ normalizes current speed by the maximum allowed $v_{\text{max}}$, and termination reward $r_i^{\text{termination}} = c_{\text{goal}}$ if goal reached (if collision or out-of-lane, $r_i^{\text{termination}} = c_{\text{fail}}$). $c_{\text{goal}}$ is always positive, and $c_{\text{fail}}$ is set as a negative.

To encourage cooperation, we further define a cooperative reward $r_i^C = \sum_{j \in \mathcal{N}_i} r_j^I$, where $\mathcal{N}_i$ is the set of vehicles within communication range. The total reward is then $r_i = (1 - \lambda_{\text{co}})\, r_i^I\ +\ \lambda_{\text{co}}\, r_i^C$, balancing individual performance ($\lambda_{\text{co}} = 0$) and team coordination ($\lambda_{\text{co}} > 0$).

### E.1.2   SCENARIO DESCRIPTIONS

We provide two types of driving scenarios, intersection and parking lot. These environments are designed to test the agent's communication ability in large-scale MARL and the holistic coordination ability of their action, considering safety, goal achieving, and agility.

**Intersection:** An unprotected four-way intersection without traffic signals. Agents must negotiate right-of-way, decide when to turn or go straight, and avoid gridlock. Explicit communication of intentions (e.g. yield, turn left) reduces ambiguity and improves safety.

**Parking Lot:** A confined lot with eight parking slots and spawn points both inside and on adjacent roads. Vehicles may need to reverse, yield, or reroute to find a spot. Real-time information exchange helps agents agree on who moves when and where to park without blocking traffic.

## E.2 ROBOTARIUM

(*Limited observability, cooperative, and robotics*) The Robotarium platform is a remotely accessible multi-robot lab from Georgia Tech, enabling researchers to conduct physical robot experiments with ease. In this setup, mobile robots that do not have sensors or LiDAR rely on inter-robot communication to coordinate actions, avoid collisions, and stay within bounds. We evaluate our method in two scenarios with four robotic agents: simple navigation and material transport. In such a simple simulator, we test their coordination performance without observation devices.

**Simple Navigation:** During each episode, the swarm robots navigate toward a destination point that may vary across episodes. Each agent is provided only with its own position and the destination's coordinates as observations.

In this scenario, privileged state at time $t$ is $\hat{s}^t = \left\{ (x_i^t, y_i^t),\ (x_i^{\text{goal}}, y_i^{\text{goal}}) \right\}_{i=1}^{N}$, where $(x_i^t, y_i^t)$ is agent $i$'s position and $(x_i^{\text{goal}}, y_i^{\text{goal}})$ is its fixed goal location. Each agent $i$ receives a local observation $o_i^t = \left[ x_i^t,\ y_i^t,\ x_i^{\text{goal}},\ y_i^{\text{goal}} \right] \in \mathbb{R}^4$, containing its own coordinates and the coordinates of its goal. After receiving $o_i^t$, Agents choose from five discrete actions $\mathcal{A} = \{\text{left, right, up, down, no\_action}\}$. A no_action leaves the agent in place, and other actions move the agent by a fixed distance $d_{\text{step}}$ in the corresponding cardinal direction.

Subsequently, at each timestep, agent $i$ receives a mixed reward as an outcome of a joint action $r_i^t = (1 - \lambda_c o)\, r_i^I + \lambda_c o\, r_i^G$, where $r_i^I = -\left\| (x_i^t, y_i^t) - (x_i^{\text{goal}}, y_i^{\text{goal}}) \right\|^2$, $r_i^G = \frac{1}{N-1} \sum_{j \neq i} r_j^I$. An error penalty of $-5$ is applied if agent $i$ collides with another agent or violates workspace boundaries.

**Material Transport:** In this scenario, there are two types of swarm robots: slow and fast robots. They aim to move loads from the loading zones to the unloading zone (target). In particular, Figure 6 (g) shows that loading and unloading zones are colored purple and green; orange and red circled robots are slow and fast ones, respectively. To enhance efficiency, these robots need inter-agent communication to coordinate their behaviors for avoiding collision and splitting their workstations (*i.e.,* fast robots move the material from a distant loading area, and slow robots work in close one.

We designed MDP for such a scenario as follows. First, a privileged state at time $t$ is $\hat{s}^t = \left[ \left\{ (x_i^t, y_i^t),\ l_i^t, \tau_i^t, v_i^t \right\}_{i=1}^{N},\ z_1^t,\ z_2^t \right]$, where $(x_i^t, y_i^t)$ is agent $i$'s position, $l_i^t$ is its current load, $\tau_i^t$ is its torque, $v_i^t$ is its speed, and $z_1^t, z_2^t$ are the remaining loads at zone 1 and zone 2. Each agent $i$ can get an observation $o_i^t = \left[ x_i^t,\ y_i^t,\ l_i^t,\ z_1^t,\ z_2^t \right] \in \mathbb{R}^5$. Agents make a decision within the same action space with the simple navigation scenario. At each timestep, agent $i$ receives the reward as follows:

$$r_i^t = c_{\text{step}} + c_{\text{load-close}}(l_i^t m_{\text{load}}) + c_{\text{load-distant}}(l_i^t m_{\text{load}}) + + c_{\text{unload}}(l_i^t m_{\text{unload}}) + c_{\text{safety}} \mathbb{1}_{\text{safety}},$$

where $c_{\text{step}}$ is the constant time penalty per step, $c_{\text{load-close}}$ and $c_{\text{load-distant}}$ are the reward coefficient for loading at zone 1 and 2, respectively. Next, $c_{\text{unload}}$ is the reward coefficient for unloading at the target area, $c_{\text{safety}}$ is the penalty coefficient for safety violations (collision or boundary exit), and $\mathbb{1}_{\text{safety}}$ is an indicator function equal to 1 if a safety event occurs, 0 otherwise. Same with simple navigation, we use a combination of individual and collaborative rewards to enhance coordination.

### E.3 MULTI-AGENT QUADRUPED ENVIRONMENTS

(*Realistic, limited observability, high-dimensional observation space, cooperative, and robotics*)
MQE is an Isaac Gym–based simulator that marries physically accurate rigid-body dynamics with
massively parallel GPU execution. Each robot, Unitree Go1, is modelled with 12–18 actuated degrees
of freedom, ground–contact friction, and joint-space torque limits, enabling realistic behaviors such
as trotting, bounding, and recovery from perturbations. The platform natively supports heterogeneous
morphologies, randomized terrain tiles, and object manipulation, making it ideal for testing coordina-
tion and communication under high-dimensional, contact-rich dynamics. In this testbed, we consider
three scenarios: Go1Gate, Go1Sheep-Hard, and Go1Seesaw.

**Go1Gate:** Two Unitree Go1 robots must pass through a constricted opening of width $w_{\text{gate}}$ and
reach a specified goal region on the opposite side within a horizon of $T_{\max}$ timesteps, all without
colliding with the gate boundaries and another robot. Success demands precise alignment of the
robot's heading and finely tuned speed control to negotiate the narrow aperture; any contact incurs a
collision penalty, while a clean, collision-free traversal earns a completion bonus. This task, therefore,
emphasizes agile steering, dynamic balance, and minimal lateral deviation from the gate centerline to
ensure efficient and safe passage.

For training with privileged information at each timestep, agent $i$ has access to

$$s^i = \Big[\underbrace{\mathbf{p}_i}_{6}, \underbrace{\mathbf{p}_j}_{6}, \underbrace{\mathbf{g}}_{2}, \underbrace{\mathbf{q}_i}_{4}, \underbrace{\mathbf{v}_i}_{3}, \underbrace{\boldsymbol{\omega}_i}_{3}, \underbrace{\mathbf{d}_i}_{12}, \underbrace{\dot{\mathbf{d}}_i}_{12}, \underbrace{\mathbf{a}_i^{\text{last}}}_{12}\Big],$$

where $\mathbf{p}_i = [x_i, y_i, z_i, \text{roll}_i, \text{pitch}_i, \text{yaw}_i] \in \mathbb{R}^6$ is the concatenation of agent $i$'s base position
and orientation (Euler angles), $\mathbf{p}_j \in \mathbb{R}^6$ is the same base position and orientation of the other
agent $j \neq i$ (obtained by flipping $\mathbf{p}_i$ in the batch), $\mathbf{g} = [g_x, g_y] \in \mathbb{R}^2$ is the 2D gate position,
$\mathbf{q}_i = [q_w, q_x, q_y, q_z] \in \mathbb{R}^4$ is the agent's base orientation as a unit quaternion, $\mathbf{v}_i = [v_x, v_y, v_z] \in \mathbb{R}^3$
is the base linear velocity, $\boldsymbol{\omega}_i = [\omega_x, \omega_y, \omega_z] \in \mathbb{R}^3$ is the base angular velocity, $\mathbf{d}_i \in \mathbb{R}^{12}$ and
$\dot{\mathbf{d}}_i \in \mathbb{R}^{12}$ are the per-joint positions and velocities for the 12 DoFs, and $\mathbf{a}_i^{\text{last}}$ is the previous action
(torque or target position) applied at each joint.

At each timestep, agent $i$'s local observation is a part of privileged information, $o_i = [\mathbf{p}_i, \mathbf{p}_j, \mathbf{g}]$. An
agent $i$ outputs an action $a_i = [v_x, v_y, \omega_{\text{yaw}},]$, where $v_x, v_y$ are the desired linear velocities in the
$x, y$ directions and $\omega_{\text{yaw}}$ is the desired yaw rate. After performing the action, the agent $i$ receives a
reward as follows.

$$r_i^t = \underbrace{c_{\text{target}} \left(d_i^{t-1} - d_i^t\right)}_{\text{approach reward}} + \underbrace{c_{\text{col}} \, \mathbb{1}_{\text{col}}}_{\text{collision punishment}} + \underbrace{c_{\text{succ}} \, \mathbb{1}\big[x_i > d_{\text{gate}} + 0.25\big]}_{\text{Goal achievement reward}} + \underbrace{c_{\text{agent}} \, \tfrac{\mathbb{1}[d_{ij} < \delta]}{d_{ij}}}_{\text{Inter-agent distance punishment}}$$

**(I)** Approach reward $r_i^{\text{approach}}$ encourages agent $i$ to reduce its Euclidean distance to the target
$d_i^t = \big\|[x_i^t, y_i^t] - \text{target}\big\|_2$. The term $d_i^{t-1} - d_i^t$ is positive when the agent moves closer, and $c_{\text{target}}$
scales the reward magnitude. **(II)** Collision punishment $r_i^{\text{contact}}$ imposes a penalty $c_{\text{col}}$ whenever
agent $i$ collides, indicated by $\mathbb{1}_{\text{col}} = 1$. This discourages unsafe actions. **(III)** Goal achievement
reward $r_i^{\text{success}}$ awards a one-time bonus $c_{\text{succ}}$ when agent $i$ first crosses the gate threshold at
$x_i > d_{\text{gate}} + 0.25$, as marked by the indicator. **(IV)** Inter-agent distance punishment $r_i^{\text{agent}}$ discourages
agents from crowding by penalizing when the squared inter-agent distance $d_{ij}^2 = \|[x_i, y_i] - [x_j, y_j]\|_2^2$
is below $\delta$. The penalty $c_{\text{agent}}/d_{ij}^2$ grows as they get closer.

**Go1Sheep-Hard:** In this task, two Go1 robots must guide nine simulated sheep NPCs from their
initial spawn area into a designated corral within a fixed horizon of $T_{\max}$ timesteps, strictly avoiding
collisions. Each sheep executes randomized maneuvers and unpredictable bursts of acceleration,
demanding that the robot continuously predict their future positions and adjust its steering and speed
with high agility. Task success hinges on efficiently reducing the Euclidean distance between the
robot and each sheep to enable timely interception, steering the flock toward the corral boundary, and
preserving a safe buffer to prevent contact while the sheep actively attempt to escape capture.

For Go1Sheep MDP, privileged information first is defined as follows:

$$s^i = \Big[\mathbf{p}_i, \mathbf{p}_j, \mathbf{g}, \underbrace{\mathbf{m}}_{2 \times 9}, \mathbf{q}_i, \mathbf{v}_i, \boldsymbol{\omega}_i, \mathbf{d}_i, \dot{\mathbf{d}}_i, \mathbf{a}_i^{\text{last}}\Big],$$

where $\mathbf{m} = [x_{\mathrm{npc1}}, y_{\mathrm{npc1}}, \ldots, x_{\mathrm{npc9}}, y_{\mathrm{npc9}}]$ are the 2D positions of the nine 'sheep' NPCs. The observation is $o_i = [\mathbf{p}_i, \mathbf{p}_j, \mathbf{g}, \mathbf{m}]$, and action is same with Go1Gate. The reward is defined as follows.

$$
r_i^t = \underbrace{c_{\mathrm{succ}} \sum_{m=1}^{M} \mathbb{1}\left[x_{s_m}^t > d_{\mathrm{gate}}\right]}_{\text{Success shepherding reward}} + \underbrace{\begin{cases} c_{\mathrm{mix}}, & x_{s_m}^t \geq d_{\mathrm{gate}}, \\ c_{\mathrm{mix}} \ \exp\left(-\dfrac{\|[x_{s_m}^t, y_{s_m}^t] - \mathbf{g}\|}{2}\right), & \text{otherwise} \end{cases}}_{\text{Mixed sheep reward}}
$$

**(I)** Success shepherding reward $r_i^{\mathrm{success}}$ encourages the agent to drive all NPCs across the gate. Let $\mathrm{cross}_{e,m} = \mathbb{1}\left[x_{s_e,m}^t > d_{\mathrm{gate}}\right]$, $S = \sum_{m=1}^9 \mathrm{cross}_m$. Then each agent in environment receives $r_i^{\mathrm{success}} = c_{\mathrm{succ}} \ S,$. This one-time shaping reward is added each timestep after any sheep crosses the gate, and is reset when the environment resets. **(II)** Mixed sheep reward $r_i^{\mathrm{mixed}}$ provides a continuous shaping signal based on each sheep's proximity to the gate. In this shaping scheme, we first compute the two-dimensional Euclidean distance $D = \|(x,y) - \mathbf{g}\|_2$ between each sheep's position $(x, y)$ and the gate center $\mathbf{g}$. The per-sheep reward $r_{\mathrm{mixed}}$ is then defined as $r_i^{\mathrm{mixed}} = c_{\mathrm{mix}}$ when $x \geq d_{\mathrm{gate}}$, otherwise, $r_i^{\mathrm{mixed}} = c_{\mathrm{mix}} \exp(-D/2)$, so that once a sheep crosses the gate threshold $d_{\mathrm{gate}}$ it immediately earns the full shaping scale $c_{\mathrm{mix}}$, while before crossing it receives a smoothly increasing bonus that decays exponentially with distance. Summing these per-sheep terms over all sheep produces the total mixed-sheep reward, which therefore rises continuously as the flock approaches the gate and caps at $c_{\mathrm{mix}}$ upon passage.

**Go1Seesaw:** Two Unitree Go1 quadrupeds must coordinate to exploit a lever-style plank and climb onto an adjacent elevated platform within a maximum of $T_{\mathrm{max}}$ timesteps, without causing the seesaw to collapse. Starting on one side of a suspended flat board, one robot must position itself at the far end to counterbalance and stabilize the seesaw's pivot, while the second robot ascends the inclined plank to reach the platform. Precise timing of weight distribution, agile modulation of stance to damp oscillations, and real-time adaptation to shifting center-of-mass dynamics are all required to maintain balance and complete the ascent successfully.

In this task, privileged information is defined as $s^i = \left[\mathbf{p}_i, \mathbf{p}_j, \mathbf{q}_i, \mathbf{v}_i, \boldsymbol{\omega}_i, \mathbf{d}_i, \dot{\mathbf{d}}_i, \mathbf{a}_i^{\mathrm{last}}\right]$, the observation $o^i = \left[\mathbf{p}_i, \mathbf{p}_j\right]$, and action is same with other tasks. Lastly, the reward is defined as follows.

$$
r_i^t = \underbrace{c_x \left(\sum_{i=1}^2 x_i^t - \sum_{i=1}^2 x_i^{t-1}\right)}_{x-\text{movement reward}} + \underbrace{c_h \left(\sum_{i=1}^2 z_i^t - 0.56N\right)}_{\text{Height reward}} + \underbrace{c_y \left(\sum_{i=1}^2 (y_i^t)^2 - 0.5N\right)}_{\text{Lateral-deviation punishment}} +
$$

$$
c_{\mathrm{col}} \ \mathbb{1}_{\mathrm{col}}^t + \underbrace{\frac{c_{\mathrm{dist}} \ \mathbb{1}[d_{ij}^t < 0.5]}{(d_{ij}^t)^2}}_{\text{Inter-agent distance punishment}} + \underbrace{c_{\mathrm{succ}} \sum_{i=1}^2 \mathbb{1}\left[x_i^t > 7.7 \ \wedge \ z_i^t > 1.3\right]}_{\text{Goal achievement reward}} + \underbrace{c_{\mathrm{fall}} \ \mathbb{1}_{\mathrm{fall}}^t}_{\text{fall punishment}}
$$

**(I)** x-movement reward $r^{\mathrm{move}}$ encourages the robots to collectively advance along the plank by rewarding positive change in their fore–aft positions. **(II)** Height reward $r^{\mathrm{height}}$ incentivizes climbing by granting a bonus whenever the team's average elevation exceeds the nominal stance height. In this term, $Z$-coordinates above the nominal standing height (0.56m) earn a bonus. **(III)** Lateral-deviation punishment $r^y$ penalizes straying from the centerline by imposing a cost for large side-to-side displacements, thereby keeping the bases near the seesaw's centerline. **(IV)** Collision punishment $r^{\mathrm{col}}$ discourages unsafe contacts by applying a fixed penalty whenever any robot registers an external collision, promoting safe foot placements and mutual avoidance. **(V)** Inter-agent distance punishment $r^{\mathrm{dist}}$ prevents crowding by sharply penalizing pairs of robots whose planar separation drops below 0.5m. **(VI)** Goal achievement reward $r^{\mathrm{succ}}$ rewards task completion by giving each robot a one-time bonus when it steps cleanly onto the far platform ($x > 7.7$ m and $z > 1.3$ m). **(VII)** Fall punishment $r^{\mathrm{fall}}$ strongly discourages loss of balance by penalizing any roll- or pitch-termination event.

## E.4 BI-DEXHANDS

(*Realistic, limited observability, high-dimensional observation space, high-dimensional action space, cooperative, and robotics*) Bi-DexHands is a high-fidelity bimanual manipulation benchmark built in Isaac Gym, pairing two Shadow Hands in richly contact-driven tasks. In this work, we focus on three prototypical scenarios, *i.e.*, Door Open Outward, Bottle Cap, and Two Catch Underarm, each stressing different aspects of coordinated wrist and finger control. We select this testbed to push RL toward human-level dexterity for several reasons: Isaac Gym's GPU parallelism delivers massive sample efficiency; the dual-hand setup embodies heterogeneous-agent cooperation in a very high-dimensional action space; and the tasks themselves are grounded in cognitive studies of fine-motor skill development, enabling evaluation of skill acquisition stages.

**Door Open Outward:** In this bimanual lever task, one Shadow Hand must firmly grasp a hinged door handle while the other pushes the door open away from the robot, reaching a target angle before $T_{\max}$ timesteps, without dropping the handle or colliding with the door frame. Success demands that the "grasp" hand maintain a stable closure force as the "push" hand applies a sustained outward force and trajectory, balancing leverage and support. This challenge, therefore, stresses persistent contact stability, force distribution between hands, and dynamic coordination to overcome hinge resistance. From the viewpoint of human development, this task can be performed after 13 months (Weiss et al., 2010).

Privileged information for Door Open Outward is defined as follows:

$$s^t \in \mathbb{R}^{428} = \left[ s^{\mathrm{RH},t},\ s^{\mathrm{LH},t},\ s^{\mathrm{door},t} \right],$$

where the superscripts "RH" and "LH" denote the Right and Left Hand blocks (each of dimension 199, total 398), and "door" the 30-dimensional door/goal block.

In particular, each 199-dimensional hand block $s^{\mathrm{H},t}$ (for $\mathrm{H} \in \{\mathrm{RH}, \mathrm{LH}\}$) we further decompose as follows. Joint positions $\left[ s_{0:23}^{\mathrm{H},t} \right] \in \mathbb{R}^{24}$, Joint velocities $\left[ s_{24:47}^{\mathrm{H},t} \right] \in \mathbb{R}^{24}$, Joint efforts $\left[ s_{48:71}^{\mathrm{H},t} \right] \in \mathbb{R}^{24}$, Fingertip kinematics (positions, linear and angular velocities) $\left[ s_{72:136}^{\mathrm{H},t} \right] \in \mathbb{R}^{65}$, Fingertip forces and torques $\left[ s_{137:166}^{\mathrm{H},t} \right] \in \mathbb{R}^{30}$, Base position $\left[ s_{167:169}^{\mathrm{H},t} \right] \in \mathbb{R}^{3}$, Base orientation (roll, pitch, yaw) $\left[ s_{170:172}^{\mathrm{H},t} \right] \in \mathbb{R}^{3}$, Previous action commands $\left[ s_{173:198}^{\mathrm{H},t} \right] \in \mathbb{R}^{26}$. The remaining 30 dimensions $s^{\mathrm{door},t}$ encode the door's pose (7), linear velocity (3), angular velocity (3), goal pose (7), and the right/left handle positions $(3 + 3)$, in that order.

Each hand gets its own hand and object information as a local observation at a timestep $t$.

Next, the action at time $t$ is a 26-dimensional continuous vector for each hand, $a^t = \left[ a_{0:25}^t \right] \in \mathbb{R}^{26}$, which we partition into six contiguous blocks:

$$a^t = \left[\ \underbrace{a_{0:19}^{\mathrm{RH}}}_{\substack{\text{Hand} \\ \text{joint commands}}}\ ,\ \underbrace{a_{20:22}^{\mathrm{RH}}}_{\substack{\text{Hand} \\ \text{base pose}}},\ \underbrace{a_{23:25}^{\mathrm{RH}}}_{\substack{\text{Hand} \\ \text{orientations}}}\ \right].$$

In particular, $a_{0:19}^{\mathrm{H}}$ and $a_{26:45}^{\mathrm{LH}}$ are the 20 per-hand joint actuator targets, and $a_{20:22}^{\mathrm{H}}$, $a_{46:48}^{\mathrm{LH}}$ are the palm translations $(x, y, z)$, and $a_{23:25}^{\mathrm{H}}$, $a_{49:51}^{\mathrm{LH}}$ are the palm orientations (roll, pitch, yaw).

At every step, the agent earns

$$r = 0.2\ -\ \|x_{\ell\mathrm{hand}} - x_{\ell\mathrm{handle}}\|_2\ -\ \|x_{r\mathrm{hand}} - x_{r\mathrm{handle}}\|_2\ +\ 2\,\|x_{\ell\mathrm{handle}} - x_{r\mathrm{handle}}\|_2.$$

The two negative terms penalize any drift from each handle, ensuring firm contact, while the final term encourages opening by increasing the handle-to-handle separation.

**Open Bottle Cap**: The dual Shadow Hands must grasp a bottle body and its screw-on cap, then unscrew the cap outward by applying controlled counter-rotational torque, all within $T_{\max}$ timesteps and without slip or excessive force. One hand holds the bottle steady while the other hand applies a precise twisting motion to overcome the cap's friction; any loss of grip or collision with the bottle incurs a penalty, whereas a clean uncapping within the time limit yields a completion bonus. This scenario highlights coordinated torque control, compliant grip modulation, and synchronized wrist rotation. From the viewpoint of human development, this task can be performed after 30 months (Zubler et al., 2022).

Privileged information for Open Bottle Cap is defined as follows:

$$s^t \in \mathbb{R}^{417} = \left[ s^{\mathrm{RH},t}, \ s^{\mathrm{LH},t}, \ s^{\mathrm{bottle},t}, \ s^{\mathrm{cap},t} \right],$$

where $s^{\mathrm{bottle},t}$ and $s^{\mathrm{cap},t}$ is information related to task objects. First, $s^{\mathrm{bottle},t}$ includes bottle pose (position and orientation quaternion; 7), linear velocity (3), and angular velocity (3). Next, $s^{\mathrm{cap},t}$ can be decomposed as cap position (3) and cap up vector (3).

The reward combines hand-to-object proximity penalties with a strong shaping term for cap separation:

$$r = 0.2 \ - \ \|x_{\ell \mathrm{hand}} - x_{\mathrm{cap}}\|_2 \ - \ \|x_{r \mathrm{hand}} - x_{\mathrm{bottle}}\|_2 \ + \ 30 \, \|x_{\mathrm{bottle}} - x_{\mathrm{cap}}\|_2.$$

Here the first two negative terms discourage loss of contact, one hand holding the bottle body and the other gripping the cap, while the large coefficient on $\|x_{\mathrm{bottle}} - x_{\mathrm{cap}}\|$ sharply rewards successful unscrewing by maximizing the distance between bottle and cap.

**Two Catch Underarm:** Two Shadow Hands must cooperatively toss and catch two rigid objects underarm within a fixed horizon of $T_{\mathrm{max}}$ timesteps, without dropping either object. Success requires precisely timed wrist and finger trajectories to launch each object into free flight and intercept it in the opposite palm, while maintaining stable hand poses to avoid collision between the hands and the objects. This task, therefore, emphasizes accurate throw, catch timing, trajectory prediction, and rapid coordination of dual manipulators under gravity. From the viewpoint of human development, this task can be performed after becoming an adult.

Privileged information for Two Catch Underarm is defined as follows:

$$s^t \in \mathbb{R}^{417} = \left[ s^{\mathrm{RH},t}, \ s^{\mathrm{LH},t}, \ s^{\mathrm{object}\ 1,t}, \ s^{\mathrm{object}\ 2,t} \right],$$

where $s^{\mathrm{object},t}$ comprises of object pose (position and orientation quaternion; 7), linear velocity (3), angular velocity (3), goal pose (desired pose; 7), and rotational error between object and goal (4).

At each timestep, the agent receives a reward that sums two exponential pose-error terms, one for each object:

$$r = \exp\left[-0.2\left(\alpha \, d_{t1} + d_{r1}\right)\right] \ + \ \exp\left[-0.2\left(\alpha \, d_{t2} + d_{r2}\right)\right],$$

where $d_{ti} = \|x_{o_i} - x_{g_i}\|_2$ is the Euclidean distance between object $i$ and its goal, and $d_{r_i} = 2 \arcsin\left(\mathrm{clamp}(\|da_i\|_2, 1)\right)$ measures the rotational misalignment. The dual-exponential form sharply penalizes both translational and rotational errors for each throw-catch pair, driving the hands to synchronize their toss and intercept trajectories.

## E.5 HYPERPARAMETERS

| Hyperparameter | Value |
|---|---|
| Max steps of an episode | 1000 (MetaDrive), 50 (SN), 60 (MT), 200 (MQE), 400 (BiDexHands) |
| Total timesteps | $10^6$ (MetaDrive), $2 \times 10^5$ (Robotarium), $10^7$ (MQE, BiDexHands) |
| The number of multi-threads | 16 (MetaDrive), 32 (Robotarium), 250 (MQE and BiDexHands) |
| Batch size | num threads $\times$ buffer length $\times$ num agents |
| Mini batch size | batch size / mini-batch |
| Dimensions for communication processor | 128 |
| Dimensions for latent | 32 |
| Balancing coefficient of interactive loss | 0.05 |
| Balancing coefficient of world loss | 0.05 |
| The number of heads for communication processor | 4 |
| Commitment coefficient | 0.1 |
| Value loss | Huber loss |
| Threshold of Huber loss | 10.0 |
| Recurrent data chunk length | 10 |
| Dimensions for policy | [128] |
| Dimensions for value | [128] |
| Clip coefficient of PPO | 0.2 |
| Discount factor | 0.99 |
| GAE lamda | 0.95 |
| Gradient clip norm | 10.0 |
| Optimizer epsilon | $10^{-5}$ |
| Weight decay | 0 |
| policy learning rate | $3 \times 10^{-4}$ |
| value learning rate | $3 \times 10^{-4}$ |
| Optimizer | Adam |
| RL network initialization | Orthogonal |
| Use reward normalization | True |
| Use feature normalization | True |

## E.6 BASELINE ALGORITHMS

This work benchmarks four MARL baseline algorithms. The implementation code adheres closely to the aforementioned official code as follows.

- MAPPO: https://github.com/zoeyuchao/mappo
- MAT: https://github.com/PKU-MARL/Multi-Agent-Transformer
- MAGIC: https://github.com/CORE-Robotics-Lab/MAGIC
- CommFormer: https://github.com/charleshsc/CommFormer

# F ADDITIONAL RESULTS

## F.1 ABLATION STUDY

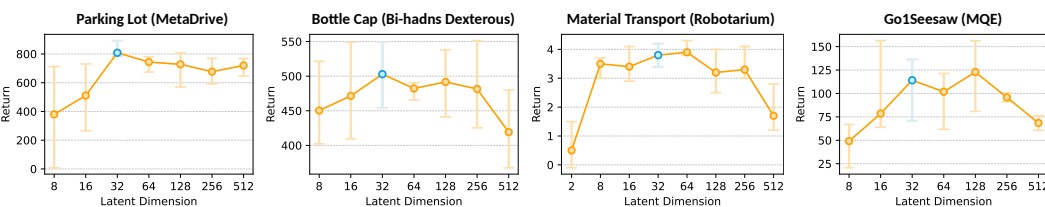

Figure 10: **Ablation study of latent dimension for MARL.** In four testbeds, $D = 32$ generally leads to the best or near-best performance.

Figure 10 plots the effect of the interaction-world latent dimension $\mathcal{Z} \in \mathbb{R}^D$ on coordination performance across four diverse tasks. We evaluate $D = \{8, 16, 32, 64, 128, 256, 512\}$ and display average return with a tolerance interval. In all environments, performance rises sharply as $D$ increases from 8 to 32, peaks or plateaus in the range $32 \le D \le 128$, and then degrades slightly at very high dimensions, suggesting that overly small latent spaces underfit inter-agent and world structure, while excessively large ones suffer from over-parameterization. Based on these results, we choose $D = 32$ for all main experiments.

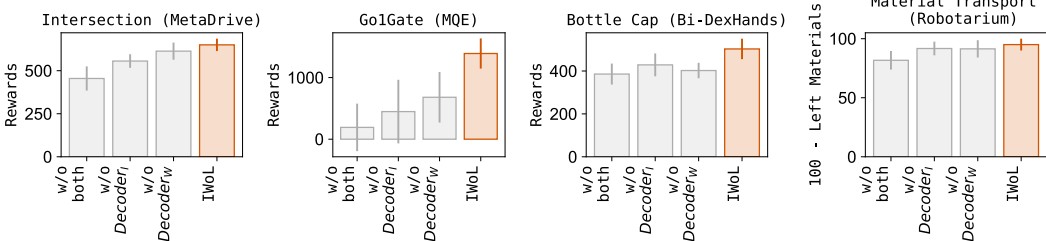

Figure 11: **Ablation study of** $\text{Decoder}_{\text{W}}$ **and** $\text{Decoder}_{\text{I}}$. Across four challenging tasks, $\text{Decoder}_{\text{W}}$ and $\text{Decoder}_{\text{I}}$ are important for multi-agent coordination.

Figure 11 shows the effect of our contributed modules, $\text{Decoder}_{\text{W}}$ and $\text{Decoder}_{\text{I}}$ on coordination performance across four tasks. This result confirms that these modules substantially influence the performance of IWoL. More precisely, in MQE and MetaDrive, removing either decoder leads to drastic reward degradation, effectively preventing coordination. In Bi-DexHands and Robotarium scenarios, IWoL with both modules achieves the most stable and highest performance. This indicates that $\text{Decoder}_{\text{W}}$ and $\text{Decoder}_{\text{I}}$ play complementary roles, the former capturing world-level dynamics and the latter modeling agent interactions, together enabling robust coordination.

## F.2 TRAINING TIME

To assess the practical training cost of IWoL, we report wall-clock times for all baselines across representative environments. In MetaDrive, Im-IWoL and Ex-IWoL require 8 and 10 hours, respectively, comparable to MAPPO (8h) and notably faster than MAT (11h), CommFormer (13h), and MAGIC (9h). In Bi-DexHands, where coordination and contact-rich manipulation increase complexity, Im-IWoL trains in 13 hours and Ex-IWoL in 17 hours faster than CommFormer (20h) and MAT (18h), and similar to MAGIC (15h), though MAPPO remains the fastest at 6 hours. A similar pattern emerges in Go1 Tasks, with IWoL variants showing $13 - 17$ hours of training time versus $15 - 20$ hours for communication-based baselines, and MAPPO again finishing in 6 hours. These results suggest that IWoL achieves strong performance with modest additional overhead compared to message-free methods, and substantially better scalability than Transformer-based communicators.

### F.3 INFERENCE TIME AT DEPLOYMENT

Figure 12 compares the average inference time for each agent in four environments across all baselines. MAPPO serves as the lightweight baseline, while MAT, with its full transformer encoder, and explicit communication baselines, requiring per-step message exchanges, both incur noticeable overhead. By contrast, Im-IWoL reuses its pretrained latent encoder and adds only a small fully-connected projection on top of the MAPPO. As a result, its runtime is virtually indistinguishable from MAPPO and substantially faster than both MAT and explicit communication, demonstrating that learned representations can boost coordination without sacrificing efficiency.

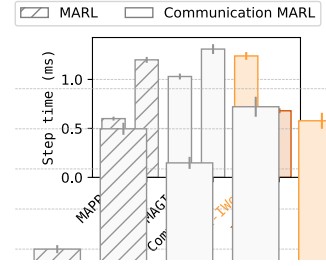

Figure 12: **Inference run time.** Wall-clock time for each agent.

### F.4 COMPARISON WITH ICP

Although we claim the differences of fundamental idea between ICP and IWoL in **RQ5**, this subsection takes into account the following research question: **How well Im-IWoL performs compared to previous Implicit Communication Protocols in robotics frameworks?**

Implicit Channel Protocol (ICP), most recent work, enables agents to communicate without a dedicated messaging channel by treating certain observable actions as encoded signals in the environment. At each time step, an agent chooses between a scouting action $u^s = \mathcal{P}(m)$ to transmit message $m$ via a predefined bijection $\mathcal{P} \colon M \to U^s$, or a regular action $u^r$ to affect the environment. Other agents observe $u^s$ in their next local observation and recover $m$ by applying the inverse mapping $\mathcal{P}^{-1}$. Through this mechanism, ICP realizes implicit communication as inverse modeling: the behavior itself becomes the communication signal.

While (Wang et al., 2025) shows that ICP achieves better performance than other branches, these empirical results in discrete extensive-form game tasks are based on some assumptions.

- **Signal broadcast:** When an agent executes $u^s$, the environment inserts that action identifier into every other agent's next observation, ensuring universal message delivery.
- **Perfect encoding/decoding:** There exists a bijective mapping $\mathcal{P}/\mathcal{P}^{-1}$ between messages $M$ and scouting actions $U^s$, allowing lossless recovery.

These premises rely on perfectly and instantaneously observing every other agent's actions, an idealization that is often violated in robotics and other real-world domains.

Consequently, we extend the ICP module to align with our experiments. First, we introduce a shared replay buffer that logs every agent's executed actions to train the encoder and decoder. In other words, the multiple agents use a homogeneous encoder and decoder module in the execution phase to decode other agents' embedded actions. Additionally, we maintain the signal broadcast assumption, that is, each agent can observe others' embedded actions as a detectable environmental factor. Lastly, we adopt the three types of value functions: fully decentralized value function (ICP-Dec), global value function as the sum of individual value function (ICP-Sum), and global value function as the monotonic mixing of individual value function (ICP-Monotonic) in the training phase.

In the main text, Table 3 shows the performance comparison between Im-IWoL and ICP variants in three tasks. First of all, Im-IWoL demonstrates clear superiority compared to ICP. Both ICP-Sum and ICP-Monotonic consistently exceed the fully decentralized baseline by leveraging structured value decomposition and state-conditioned mixing to better capture inter-agent dependencies. Monotonic mixing affords further gains in tasks demanding tight coordination, while the simpler sum-of-values approach already delivers substantial robustness; in particular, it works better than ICP-Sum in the MetaDrive where multiple agents share the environment compared to other scenarios. In contrast, the fully decentralized variant struggles to scale as complexity grows. These results underscore the critical role of principled value combination in enhancing implicit communication and cooperative behavior.

## F.5 IWoL with Off-policy Family Algorithm

In the main body, we only consider the on-policy family of algorithms. That is because they empirically provide stable and reliable training compared to the off-policy family. Additionally, all of our selected baselines are formulated as on-policy algorithms, so adhering to on-policy ensures a fair comparison and evaluation of performance.

This subsection investigates whether IWoL can be adopted within the off-policy family of algorithms.

Table 4: **IWoL's performance in off-policy algorithm.**

|  | Algorithms | | | |
|---|---|---|---|---|
| Task | MAPPO | MAGIC | MADDPG | MADDPG+IWoL |
| Intersection | $454.8_{\pm 70.2}$ | $518.3_{\pm 77.4}$ | $413.8_{\pm 188.7}$ | $506.1_{\pm 95.7}$ |
| Parking Lot | $327.4_{\pm 211.7}$ | $371.2_{\pm 14.3}$ | $311.0_{\pm 142.6}$ | $459.3_{\pm 88.2}$ |

Table 4 MADDPG+IWoL consistently improves over vanilla MADDPG, demonstrating that IWoL can also be effectively applied in off-policy settings. These results reinforce the generality and effectiveness of our communication framework, beyond the on-policy family.

## F.6 Full Training Curve

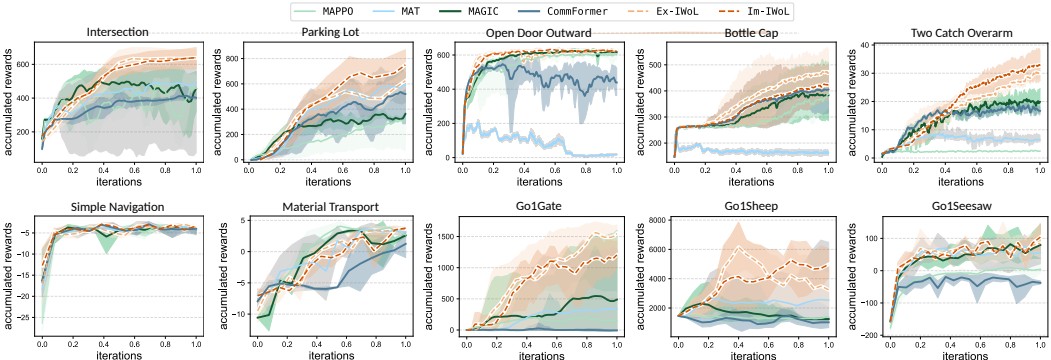

Figure 13: **Learning curves across four environments.** We plot the averaged rewards as a solid line and the tolerance interval as a shaded area.

