# OpenReview forum: "Learning to Interact in World Latent for Team Coordination"
_ICLR.cc/2026/Conference — Submitted to ICLR 2026_

### Official Review · Reviewer_eTsZ · 2025-10-28

**Soundness:** 4
**Presentation:** 3
**Contribution:** 2
**Rating:** 6
**Confidence:** 4

**Summary:**

The authors present IWoL, a unified representation learning and communication framework for cooperative multi-agent RL (MARL). The proposed method jointly learns a latent space that captures inter-agent relations and helps in addressing partial observability in MARL. The method features two variants, one with implicit communication and the other with explicit communication at execution/test time. The authors perform experiments over different scenarios/benchmarks, comparing the performance of the proposed method against other baselines.

**Strengths:**

The paper features a good discussion of related work in Sec. 2. The experimental protocol seems solid, with the authors performing multiple runs and reporting confidence intervals. The experimental results seem reproducible. The experiments are also extensive, considering several different environments. The experimental results seem to be in favour of the proposed method. The authors also provide an ablation study.

**Weaknesses:**

I think the clarity of the paper could be slightly improved in some parts, particularly in Sec. 4.2.

The proposed method seems to feature some limitations (deterministic transitions and access to the underlying state at training time).

The contributions of the paper appear somewhat incremental, as it builds upon and combines previously proposed techniques in the MARL field. Could the authors further clarify if there are key contributions (in terms of the proposed method) besides combining previously proposed techniques? Nevertheless, I believe this should not be a reason to reject the paper alone, as the paper has its own merits.

**Questions:**

- "(...) we assume a deterministic MDP throughout this work" - this means the state transition dynamics are deterministic? This is quite a restrictive assumption.
- "(...) we introduce a learnable protocol" - the learnable protocol function P needs to be trained/applied in a centralized fashion, right? I understand that for the implicit world model this is not a problem, but for the explicit world model it may be.
- line 205, "Next, the communication protocol block" -  this corresponds to the red block in Fig. 2, right? If so, please clarify it in the text. I got a bit lost while reading Sec. 4.2.
- line 214, "Lastly, the previously produced vectors, e.g., communication message and intermediate embedding, are used as input to policy and value function networks." - is this correct? Looking at Fig. 2 it seems that this is indeed the input to the value function, but the input to the policy is z_i^t. Also, z_i^t is only explained around line 244. I suggest the authors explain it earlier in the text to make the discussion clearer.
- line 247, "The world decoder reconstructs the agent’s privileged state s^t_i" - so, this assumes access to the true state of the Dec-POMDP during training? This is quite a restrictive assumption, right? What about reconstructing joint observations?
- Why would Im-IWoL outperform Ex-IWoL - E.g., looking at Fig. 7 this seems to be the case. Why? I would expect Ex-IWoL to be in advantage in comparison to Em-IWoL since it features explicit communication between the agents at execution time, right?

Minor comments/typos:
- sentence in line 17 of the abstract starting with "This representation" needs to be revised.
- sentence in line 61 "if desired, the same backbone can expose as explicit messages at test time." is a bit hard to understand.
- line 197: "by a self-attention, (...)" - "layer" missing in the sentence?

---

> ### Author Response · Authors · 2025-11-13
> **Response to reviewer's concerns (1/3)**
>
> We sincerely thank the reviewer for taking the time to read our paper with exceptional care and for providing thoughtful feedback that ranges from detailed, minor issues to clarifying major conceptual points.
>
> ---
> > **W2 & Q1** Deterministic dynamics
>
> Yes, we posit the state transition dynamics of the selected environments are deterministic. This follows common practice in prior MARL literature [1-5], especially in studies involving controlled robotic or physical simulation environments. Such an assumption simplifies analysis without loss of generality, as environmental stochasticity can be incorporated in an orthogonal manner.
>
> We agree that extending this framework to stochastic environments represents an important and promising direction for future work.
>
> ---
> > **W3** Novelty statement
>
> First of all, although the reviewer is on positive side, we would like to emphasize our technical contributions!
> - **Novel representation learning framework for MARL**
>   - **What**: IWoL, a new framework that jointly learns below two factors
>      - **Inter-agent relationship**: As multi-agent systems inherently rely on interactions and coordinated behavior, understanding which agents influence one another is essential for effective collaboration.
>      - **Privileged world information**: Using privileged information during training helps agents overcome partial observability by grounding local views, e.g.,
>         - proprioceptive states
>         - velocity, position, angle, orientation of other *observable agents*
>         - environmental physics (gravity, friction, terrain states)
>   - **Why novel**: To our knowledge, no prior MARL work explicitly disentangles and fuses these two information streams in a unified, modular encoder. This modularity lets us swap in different graph encoders or world-feature modules without retraining policies, addressing a major practical hurdle in scaling MARL.
>      - Explicit communication [1-3, 11-13]: They use explicit inter-agent relationships information without mapping into representation or integrating world information.
>      - Implicit communication [4-5]: They use explicit environmental factors, such as observable others' actions, to estimate sequential information or adapt individual behavior without explicit communication.
>      - World model or privileged information [6-7, 14-15]: They estimate the next observation, others' actions, or the global state using the learned world model.
>      - Representation learning [8-10]: They focus narrowly on specific aspects, such as local observation embedding, inter-agent relationships, or world dynamics, without integrating them.
>      - **Summary**: They require an additional module for enhancing multi-agent coordination performance at deployment, and do not investigate the entire combination like our proposed framework.
>    - **Remark**: Our solution’s another strength is its high compatibility (modularity) with other communication-based MARL solutions.
>
> - **Improvement of communication protocol**
>    - We acknowledge that each component are well-studied, but we believe that the entire framework is quite novel.
>    - **What**: We design a single communication module that can be leveraged as implicit and explicit modes.
>    - **Remarks**:
>        - A hierarchical graph generator that determines when and what to broadcast
>        - Our ablation (RQ3) shows this yields better performance vs. both pure implicit and pure explicit baselines.
>
>
> ---
> **References**
>
> [1] D. Abhishek, et al. Tarmac: Targeted multi-agent communication. ICML 2019.
>
> [2] H. Shengchao, et al. Learning Multi-Agent Communication from Graph Modeling Perspective. ICLR 2024.
>
> [3] Y. Niu et al., Multi-Agent Graph-Attention Communication and Teaming, AAMAS 2021.
>
> [4] H. Wang et al., Learning to communicate through implicit communication channels. ICLR 2024.
>
> [5] H. Wang et al., Learning to construct implicit communication channel. ICLR 2025.
>
> [6] K. Ndousse et al., Emergent social learning via multi-agent reinforcement learning. ICML 2021.
>
> [7] V. Egorov et al., Scalable multi-agent model-based reinforcement learning. AAMAS 2022.
>
> [8] P. Gu et al., Online ad hoc teamwork under partial observability. ICLR 2022.
>
> [9] D. Huh et al., Representation learning for efficient deep multi-agent reinforcement learning. CoRR 2024.
>
> [10] M. Feng et al., Joint-predictive representations for multi-agent reinforcement learning, OpenReview 2023.
>
> [11] X. Guo et al., Scalable communication for multi-agent reinforcement learning via transformer-based email mechanism. IJCAI 2023.
>
> [12] M. Wen et al., Multi-agent reinforcement learning is a sequence modeling problem. NeurIPS 2022.
>
> [13] S. Hu et al., Learning multi-agent communication from graph modeling perspective. ICLR 2024.
>
> [14] X. Li et al., Individual contributions as intrinsic exploration scaffolds for multi-agent reinforcement learning. ICML 2024.
>
> [15] K. Ndousse et al., Emergent social learning via multi-agent reinforcement learning. ICML 2021.

---

> ### Author Response · Authors · 2025-11-13
> **Response to reviewer's concerns (2/3)**
>
> > **Q2** The learnable protocol function $P$ needs to be trained/applied in a centralized fashion, right?
>
> We appreciate the reviewer’s insightful question. Yes, for Ex-IWoL, communication protocol $P$ needs to be trained/applied in a centralized fashion. On the other hand, Im-IWoL does not require $P$ in application. The learned interaction-world latent enables decentralized coordination without explicit message passing.
>
> Our work primarily focuses on building unified representation, and its product is directly Im-IWoL. Although Ex-IWoL needs to be applied in a centralized fashion, this does not contradict our main contribution; rather, it highlights the flexibility of the proposed architecture, which can accommodate both implicit (message-free) and explicit (message-rich) coordination within a unified latent representation framework.
>
> Beyond our solution, IWoL can be compared with other world models [1-3]. The most world models in multi-agent setting inherently operate in a centralized manner. Assigning each agent an independent world model would cause compounding prediction errors across agents. Alternatively, employing a shared world model distributed among agents would still require each agent to access others’ latent states. Thus, while the mechanism may appear different, it ultimately faces similar structural constraints.
>
> ---
> > **Q3** Next, the communication protocol block - this corresponds to the red block in Fig. 2
>
> We apologize for the confusion and appreciate the reviewer’s careful reading. We have clarified this point in the revised manuscript. Specifically, we now explicitly state that "the communication protocol block (colored red in Figure 2 and detailed in Figure 4) ..."
>
> ---
> > **Q4** line 214 ... and $z_i^t$ is only explained ...
>
> We thank the reviewer for pointing out this issue. We agree that the original description could be misleading. We have revised the figure to clarify that the policy network takes $z_i^t$ and $f_i^t$ as input, while the value function uses the communication message and intermediate embeddings. Moreover, we now introduce $z_i^t$​ earlier in the text to improve clarity.
>
> ---
> > **Q5** line 247... privileged state
>
> We sincerely thank the reviewer for this insightful question. We agree that directly accessing the true state would indeed be a restrictive assumption. However, the privileged state in our work does not correspond to the full underlying state of the environment. Rather, it refers to task-specific information that may include proprioceptive signals, local physical variables, or histories of actions and observations available only during centralized training.
>
> Our intention was to capture richer supervision during training. In practice, the world decoder reconstructs other agent's information in observable range and task-relevant context (e.g., steering angle, velocity, distances to lane boundaries, torque, load, and contact pose) rather than the full hidden environment state. While using privileged information is admittedly a constraint, it remains consistent with the standard CTDE paradigm widely adopted in MARL, where such partial supervision is permitted during centralized training but not required during decentralized execution.
>
> And thanks to reviewer's suggestion. Reconstructing joint observations is indeed an interesting direction. However, we found that inferring only limited observable components already yields effective latent representations for coordination. We believe that extending this line of research toward richer observation reconstruction would align more closely with theory-of-mind, inspired approaches, which is an exciting avenue for future exploration.
>
>
> ---
> **Reference**
>
> [1] Q. Zhang, et al. Centralized Model and Exploration Policy for Multi-Agent RL. AAMAS 2022.
>
> [2] Y. Zhang, et al. Decentralized Transformers with Centralized Aggregation are Sample-Efficient Multi-Agent World Models. TMLR 2025.
>
> [3] K. Tsunekawa, et al. MA-Dreamer: Coordination and communication through shared imagination. arXiv 2022.

---

> ### Author Response · Authors · 2025-11-13
> **Response to reviewer's concerns (3/3)**
>
> > **Q6** Why would Im-IWoL outperform Ex-IWoL
>
> We thank the reviewer for raising this real question. One might expect Ex-IWoL to have an inherent advantage over Im-IWoL, since explicit communication at execution time can, in principle, subsume implicit communication by simply transmitting null messages. We acknowledge this is a valid theoretical perspective.
>
> However, as shown empirically in Fig. 7 and Table 1, Im-IWoL consistently outperforms Ex-IWoL, and this performance gap reflects a fundamental difference in how the two variants construct and leverage latent representations.
>
> In Ex-IWoL, when we remove the interaction–world encoder $\mathrm{Encoder}_{\mathrm{IW}}$​, the explicit message emitted by the communication module itself is reused as the latent variable ($z_i^t = m_i^t$​). Consequently, the system focuses on transmitting compact explicit messages rather than learning a rich, integrated latent representation. Even when the agent has nothing to say, the communication protocol still initiates a message-exchange step, structurally biasing the architecture toward efficient messaging rather than world-level representation learning.
>
> In contrast, Im-IWoL eliminates explicit message passing entirely and instead learns implicit relational embeddings directly from raw observations. This allows the latent space to internalize both interaction-level and world-level information in an end-to-end manner. As we describe in lines 250–254 of the original manuscript, Im-IWoL therefore benefits from a much more expressive and holistic encoding process, which naturally leads to stronger downstream performance.
>
> We have added a short conjecture in the revised explanation of RQ1: we believe the observed performance advantage arises not from communication efficiency itself, but from the richer, jointly learned world encoding enabled by Im-IWoL, which Ex-IWoL cannot replicate due to its reliance on explicit messages rather than learned latent structure.
>
> Finally, we would like to highlight the findings from RQ4. These results show that Ex-IWoL consistently exhibits better scalability than Im-IWoL, particularly as the number of agents or the complexity of the environment increases (that is, more complex dynamics). This suggests that the explicit-message structure of Ex-IWoL enables more stable performance under large-scale settings.
>
> ---
> - **Minor comments/typos**
>
> Thank you for the careful review. We have carefully addressed all the requested revisions.
> - Request to rephrase line 17
> - Request to rephrase line 61
> - Request to address Typo in line 170
>
> ---
> **We uploaded the revised manuscript.** If there are any additional questions or points that require clarification, we would be more than happy to address them. Thank you again for your time and constructive feedback.

---

> > ### Comment · Reviewer_eTsZ · 2025-11-23
> >
> > I thank the authors for their response, which addressed and clarified my concerns.

---

> ### Author Response · Authors · 2025-11-24
>
> Once again, we deeply appreciate the insightful comments and suggestions! Should there be any leftover questions, please let us know and we will make every effort to address them during the subsequent discussion period.

---

### Official Review · Reviewer_5Kvq · 2025-10-29

**Soundness:** 2
**Presentation:** 2
**Contribution:** 2
**Rating:** 2
**Confidence:** 4

**Summary:**

The paper develops a technique to learn inter-agent communication for cooperative MARL tasks with a Transformer-based communication architecture and an encoding-decoding loss.
The technique requires inter-agent communication during training (centralized training) and is flexible to turn on (“explicit”) or off (“implicit ”) the communication during execution.

**Strengths:**

Overall, the paper is well-written with clarity.
The empirical study is systematic and presented in an organized way.

**Weaknesses:**

1. Some key structural assumptions on top of Dec-POMDPs are not explicitly state:
    - Section 3 introduces the standard Dec-POMDP formulation but additional structures are needed in Section 4, such as agent position and minimum distance for communication (line 236).
    - Even, agent "privileged state” (line 247) is critical for training but not defined.

2. The novelty of the proposed method, IWoL, is fairly limited.
    - Prior works did the learning of communication topology (e.g.,MAGIC,I2C) and used Transformer-based communication modules (CommFormer, TarMAC).
    - The DecoderW module helps IWoL a lot according to Figure 8. However, such a module seems to be flexible to be a plug-and-play module for any existing prior method. For example, CommFormer/MAGIC in Figure 8 is much better with DecoderW than without.

3. It is unclear what the paper wants to say about implicit vs explicit communication:
    - Section 4.1 argues that explicit communication is worse than implicit, but the failure example of explicit communication is caused by bandwidth limit and message-corruption, which should equally affect implicit communication (during training).
    - In principle, explicit should subsume implicit because explicit agents can choose to send null messages. In Table 1,  Im-IWoL outperforms Ex-IWoL, and the paper does not explain this performance difference.

**Questions:**

All my concerns/questions are in the Weaknesses section.

---

> ### Author Response · Authors · 2025-11-13
> **Response to reviewer's concerns (1/4)**
>
> We appreciate the reviewer's detailed feedback and valuable suggestions for enhancing our work. In response, we describe how we have revised the paper to address the reviewer's concerns and questions.
>
> ---
> > **W1-1** Dec-POMDP formulation and additional structure
>
> We appreciate the reviewer’s careful reading. We understand that the additional elements used in our communication module might give the impression that they violate or extend beyond the standard Dec-POMDP formulation. However, this is a common design pattern in communication-based MARL studies, where the core problem remains formulated under Dec-POMDP [1-3] (or Markov game [4-7]). The communication mechanism is modeled as a separate module to describe message exchange or topology constraints [1-7].
>
> To prevent potential misunderstanding, we have revised Section 3 to explicitly state that we consider communication protocol $P$ under the Dec-POMDP framework.
>
> ---
> > **W1-2** Related to privileged state
>
> We thank the reviewer’s critical comment to improve readability of our manuscript. We agree that this concept is crucial for understanding the training process. While its exact content varies across tasks (e.g., environmental parameters, dynamics factors, other observable agents' information, proprioceptive states, etc), we acknowledge that the main text did not sufficiently convey this context.
>
> To clarify, the privileged state refers to task-specific information accessible only during training, such as proprioceptive feedback, historical action trajectories, other agents’ full internal states, and underlying physical parameters. These elements serve as supervision signals for learning the world-latent representation but are not observable during execution.
>
> We have revised Section 4 to include this clarification and explicitly describe how the privileged information is incorporated into the training process.
>
> ---
> > **W2** IWoL's Novelty
>
> Firstly, we would like to summarize our technical contributions.
> - **Novel representation learning framework for MARL**
>     - **What**: IWoL, a new framework that jointly learns below two factors.
>         - **Inter-agent relationship**: As multi-agent systems inherently rely on interactions and coordinated behavior, understanding which agents influence one another is essential for effective collaboration.
>         - **Privileged information**: Using privileged information during training helps agents overcome partial observability by grounding local views.
>     - **Why novel**: No prior MARL work explicitly disentangles and fuses these two information in a unified, modular encoder. This modularity lets us swap in different graph encoders or world-feature modules without retraining policies.
>         - Explicit communication [1, 4, 8]: They use explicit inter-agent relationships information without mapping into representation or integrating world information.
>         - World model or privileged information [9-10]: They estimate the next observation, others' actions, or the global state using the learned world model.
>         - Representation learning [11-13]: They focus narrowly on specific aspects, such as local observation embedding, inter-agent relationships, or world dynamics, without integrating them.
>         - **Summary**: They require an additional module for enhancing multi-agent coordination performance at deployment, and do not investigate the entire combination like our proposed framework.
>     - **Remark**: Our solution’s another strength is its high compatibility (modularity) with other communication-based MARL solutions.
>
> ---
> **References**
>
> [1] D. Abhishek, et al. Tarmac: Targeted multi-agent communication. ICML 2019.
>
> [2] X. Guo, et al. Scalable Communication for Multi-Agent Reinforcement Learning via Transformer-Based Email Mechanism. IJCAI 2023.
>
> [3] M. Wojtala, et al. MACTAS: Self-Attention-Based Module for Inter-Agent Communication in Multi-Agent Reinforcement Learning. AAMAS 2026.
>
> [4] Y. Niu, et al., Multi-Agent Graph-Attention Communication and Teaming, AAMAS 2021.
>
> [5] W. Kim, et al., Communication in Multi-Agent Reinforcement Learning: Intention Sharing, ICLR 2021.
>
> [6] D. Chen, et al. Communication-Efficient Actor-Critic Methods for Homogeneous Markov Games. ICLR 2022.
>
> [7] Z. Chen, et al. Sample and Communication-Efficient Decentralized Actor-Critic Algorithms with Finite-Time Analysis. ICML 2022.
>
> [8] H. Shengchao, et al. Learning Multi-Agent Communication from Graph Modeling Perspective. ICLR 2024.
>
> [9] K. Ndousse et al., Emergent social learning via multi-agent reinforcement learning. ICML 2021.
>
> [10] V. Egorov et al., Scalable multi-agent model-based reinforcement learning. AAMAS 2022.
>
> [11] P. Gu et al., Online ad hoc teamwork under partial observability. ICLR 2022.
>
> [12] D. Huh et al., Representation learning for efficient deep multi-agent reinforcement learning. CoRR 2024.
>
> [13] M. Feng et al., Joint-predictive representations for multi-agent reinforcement learning, OpenReview 2023.

---

> ### Author Response · Authors · 2025-11-13
> **Response to reviewer's concerns (2/4)**
>
> > **W 2-1** Comparison with baselines
>
> We appreciate the reviewer’s prior connection and for pointing out relevant literature such as MAGIC, I2C, CommFormer, and TarMAC. These studies have indeed contributed valuable insights into learning communication structures in MARL. However, our approach introduces a novel representation learning framework that departs from traditional explicit communication paradigms.
>
> We first would like to emphasize that our solution goes beyond explicit communication. Prior methods all rely on explicit message passing during inference, whereas, IWoL shifts this paradigm toward latent representation learning as implicit communication. This design provides a unified and scalable alternative to traditional communication graph optimization.
>
> Below, we summarize the key architectural distinctions between IWoL and previous methods:
>
> - Contrast to CommFormer: CommFormer use Transformer as a monolithic backbone. In other words, all functional module (e.g., value, policy, communication and graph reasoning) are fused within a single large Transformer encoder–decoder architecture. On the other hand, our Transformer is dedicated solely to the communication protocol, not to the entire MARL pipeline.
> - Contrast to TarMAC: They propose targeted communication using _attention weights_ as targeting probabilities via single attention layer, where agents broadcast and receive messages selectively. Our solution can generalizes TarMAC’s targeting into a multi-layer Transformer that performs $L$ rounds of relational reasoning.
> - Contrast to MAGIC: Unlike MAGIC, which performs multi-stage graph scheduling through a series of sub-schedulers and sub-processors that iteratively refine communication topologies across rounds, IWoL simplifies this pipeline. Our model computes the Gumbel-Softmax-based topology once per timestep and directly feeds it as an attention prior into a Transformer block. This design effectively collapses the hierarchical scheduling process into a single differentiable attention-based module, enabling efficient multi-round message propagation without repeatedly invoking graph inference.
>
> ---
> > **W 2-2** Figure 8 and Novelty
>
> We appreciate the reviewer’s considerable comment. Figure 8 demonstrates that the proposed idea can flexibly enhance various communication-based MARL baselines. This observation actually supports our claim that the proposed components are modular and generalizable, enabling seamless integration with prior methods.
>
> We would like to highlight that, even without $\mathrm{Decoder}_W$, the proposed solution itself achieves comparable or superior performance compared to existing baselines (with and without world decoder), as shown in Figure 8. This implies that our Transformer-based communication protocol alone is already strong and effective, while the addition of $\mathrm{Decoder}_W$​ further boosts performance through better world representation learning.
>
> Therefore, Figure 8 collectively shows that each proposed module is individually powerful, and their combination yields complementary benefits. This validates the modularity and scalability of our framework rather than indicating over-reliance on a single component.

---

> ### Author Response · Authors · 2025-11-13
> **Response to reviewer's concerns (3/4)**
>
> > **W 3-1**.  Motivating example
>
> We thank the reviewer's thoughtful observation. Our core motivation lies in understanding and mitigating the practical vulunerbility of explicit message passing at execution phase, rather than during training time. This focus is consistent with the key insights highlighted by other reviewers.
>
> * `Reviewr nwwV`: the motivating example effectively demonstrates the fragility of explicit communication under realistic constraints
>
> * `Reviewer CW1E`: IWoL jointly captures inter-agent relations and world information, supporting both implicit and explicit coordination while avoiding drawbacks of explicit communication
>
> While we agree that robustness issues such as bandwidth limitation and message corruption could in principle emerge during training, our primary focus is on execution-time degradation, where these problems manifest in real-world coordination. Therefore, we would like to highlight our communication module can be easily extended. To address such bandwidth limitation issue, we have tested a simple variant by attaching vector-quantization for communication module. This modification enables compact and discrete message representations, which enhance robustness even when the communication bandwidth is forcibly reduced. In Table above, the Ex-IWoL + VQ variant demonstrates significantly improved resilience under Type I (off-dynamics) conditions, confirming that our framework is flexible.
>
> | Scenario                         | Normal        | Type I (8bps) | Type I (2bps) | Type II       |
> | -------------------------------- | ------------- | ------------- | ------------- | ------------- |
> | Ex-IWoL                          | $100.0\pm0.0$ |$86.0\pm11.4$ |$72.0\pm13.0$ |$44.0\pm20.7$ |
> | Ex-IWoL with Vector quantization | $100.0\pm0.0$ | $90.0\pm7.7$ |$88.0\pm8.4$ |$38.0\pm16.4$ |
>
> Specifically, in theory, we apply a lightweight vector quantization layer to discretize the continuous communication embeddings. Let the message embedding of agent $i$ at time $t$ be $m^t_i$. We maintain a learnable codebook $\mathcal{E} = \{ e_k \}^K_{k=1}$. During message generation, each continuous message is replaced by its nearest code via $\hat{m}^t_i = \operatorname{argmin}_{e_k \in \mathcal{E}} \|\| m^t_i - e_k \|\|_2^2$.
>
> To keep the operation differentiable, we adopt the following objective $\mathcal{L}_{\text{VQ}} = ||\operatorname{sg}[m_i^t] - e_k ||_2^2 + || m_i^t - \operatorname{sg}[e_k] ||_2^2$, where $\operatorname{sg}[\cdot]$ denotes stop-gradient. This process forces each agent’s communication message to occupy one of $K$ discrete slots, effectively limiting the communication bandwidth while maintaining semantic consistency.
>
> We have included this discussion in the Appendix as a concrete direction for future research.

---

> ### Author Response · Authors · 2025-11-13
> **Response to reviewer's concerns (4/4)**
>
> > **W3-2** Ex-IWoL and Im-IWoL
>
> We appreciate the reviewer’s theoretical perspective that explicit communication could, in principle, subsume implicit communication by allowing agents to send null messages. However, in practice, these two paradigms differ fundamentally in how information is encoded and propagated. In Ex-IWoL, even when no message is explicitly broadcast, the protocol still initiates message exchanges via a structured communication channel; thus, the design focuses on efficient message transmission, not latent representation learning. In contrast, Im-IWoL eliminates explicit message passing altogether and instead learns implicit relational embeddings that internalize both interaction and world-level information within the latent space.
>
> `line 250-254 (original manuscript)`
> ```
> For the implementation of the explicit IWoL (Ex-IWoL), we eliminate the interaction-world encoder $\mathrm{Encoder}_\mathrm{IW}$ and instead reuse the explicit message emitted by the communication module itself, $z_i^{t} = m_i^{t}$, so we only use the world decoder $\mathrm{Decoder}_\mathrm{W}$ for training. While this design simplifies the latent construction and leverages explicit communication, Im-IWoL instead learns a representation from raw observations, which can easily capture more integrated interaction and world features.
> ```
>
> As shown in line 250-254 in original manuscript (above excerpt), this distinction is explicitly discussed, where we describe that Ex-IWoL reuses explicit messages as latent variables, whereas Im-IWoL learns integrated representations directly from raw observations.
>
> In Ex-IWoL, when we remove the interaction–world encoder $\mathrm{Encoder}_{\mathrm{IW}}$​, the explicit message emitted by the communication module itself is reused as the latent variable ($z_i^t = m_i^t$​). Consequently, the system focuses on transmitting compact explicit messages rather than learning a rich, integrated latent representation. Even when the agent has nothing to say, the communication protocol still initiates a message-exchange step, structurally biasing the architecture toward efficient messaging rather than world-level representation learning.
>
> In contrast, Im-IWoL eliminates explicit message passing entirely and instead learns implicit relational embeddings directly from raw observations. This allows the latent space to internalize both interaction-level and world-level information in an end-to-end manner. As we describe in lines 250–254 of the original manuscript, Im-IWoL therefore benefits from a much more expressive and holistic encoding process, which naturally leads to stronger downstream performance.
>
> We have added a short conjecture in the revised explanation of RQ1: we believe the observed performance advantage arises not from communication efficiency itself, but from the richer, jointly learned world encoding enabled by Im-IWoL, which Ex-IWoL cannot replicate due to its reliance on explicit messages rather than learned latent structure.
>
> Finally, we would like to highlight the findings from RQ4. These results show that Ex-IWoL consistently exhibits better scalability than Im-IWoL, particularly as the number of agents or the complexity of the environment increases (that is, more complex dynamics). This suggests that the explicit-message structure of Ex-IWoL enables more stable performance under large-scale settings.
>
> ---
> **We uploaded the revised manuscript.** Should you have any additional questions or require further clarification, we would be pleased to provide it. Thank you once again for your insightful and constructive comments.

---

### Official Review · Reviewer_CW1E · 2025-11-01

**Soundness:** 3
**Presentation:** 3
**Contribution:** 2
**Rating:** 4
**Confidence:** 3

**Summary:**

This paper proposes a novel representation learning framework called Interactive World Latent (IWoL) to enhance team coordination in multi-agent reinforcement learning (MARL), which captures both inter-agent relations and task-specific world information while supporting both implicit (message-free) and explicit (message-rich) coordination modes without additional modules. Across four MARL benchmarks, IWoL variants outperform existing MARL baselines, showing strong robustness under incomplete observations and good scalability with increasing agent numbers.

**Strengths:**

1. The paper proposes the IWoL framework that jointly captures inter-agent relations and task-specific world information, supporting both implicit (message-free) and explicit (message-rich) coordination modes, avoiding drawbacks of traditional explicit communication
2. IWoL outperforms existing MARL baselines across four benchmarks with higher success rates or rewards and also show good scalability.

**Weaknesses:**

1. The implicit mode (Im-IWoL) relies on a well-trained latent representation, and its performance may degrade if the training data fails to cover diverse scenarios.
2. Compared to the simple MARL baseline (MAPPO), IWoL still has slightly higher training overhead, especially in complex tasks like bimanual dexterous manipulation.
3. The selected baselines are not comprehensive compared to what the authors listed in the related work.

**Questions:**

1. Can the authors show the performance degradation of Ex-IWoL similar to Figure 1?
2. Can you explain why implicit world latent is needed if the agents do not exchange messages during execution, and how can it keep generalizable?

---

> ### Author Response · Authors · 2025-11-13
> **Response to reviewer's concerns (1/3)**
>
> We appreciate the reviewer's detailed feedback and valuable suggestions for enhancing our work. In response, we describe how we have revised the paper to address the reviewer's concerns and questions.
>
> ---
> > **W1** relies on a well-trained latent representation
>
> We appreciate the reviewer's insightful observation and fully agree with this point. The proposed solution requires enough training time for well-trained latent representation, and its performance could degrade if the training data lacks sufficient diversity. However, this limitation is a broader and fundamental challenge faced by the MARL community. We have already witnessed remarkable progress in representation learning through the success of large-scale models, e.g., GPT and vision-language models. We believe that extending such representational capability toward multi-agent learning is a necessary and promising next step for the field.
>
> As discussed in Appendix B, we also view this as an important direction for future research. Specifically, we plan to explore adaptation and open-ended structure of the world-latent representation for further enhancing generalization under unseen scenarios. We strongly believe that advancing toward generalizable representation is not only critical for improving Im-IWoL but also represents a necessary direction for the broader MARL community to pursue.
>
> Finally, we would like to highlight our experimental results for small piece of generalization in terms of scalability. Our RQ4 in Experiment section partially includes such an analysis. Even when the environment configuration is extended beyond the training distribution, Im-IWoL maintains stable performance without significant degradation. This suggests that the proposed solution can generalize effectively to moderately novel situations, although limited.
>
> In recognition of the importance of this discussion, we have incorporated an additional note in Appendix B (Future Direction) at the end of the paper to explicitly highlight this avenue for further research.
>
> ---
> > **W2** IWoL still has slightly higher training overhead
>
> We appreciate the reviewer's thorough review and acknowledge that training efficiency is an important aspect for practicality. Indeed, IWoL introduces slightly higher training overhead compared to simple MARL baselines such as MAPPO. However, we would like to emphasize that our work primarily focuses on improving interaction reasoning and coordination performance, rather than optimizing training speed. **As shown in Appendix F.3, IWoL achieves comparable inference-time efficiency while providing significant performance gains in multi-agent cooperation**.
>
> Additionally, we would like to highlight that MAPPO represents a conventional paradigm, which is performed without communication or other additional modules. In our experiments, IWoL provides substantial coordination-driven performance gains over MAPPO, i.e., about $200$ to $480$ higher rewards in MetaDrive tasks and even $\times10$ performance gains in challenging quadruped tasks such as Go1Gate. These improvements arise from IWoL’s ability to model interaction information and global privileged informations, and we believe that **the modest additional training cost is a reasonable trade-off for achieving significantly stronger, more interpretable, and more robust multi-agent coordination**.

---

> ### Author Response · Authors · 2025-11-13
> **Response to reviewer's concerns (2/3)**
>
> > **W3**  not comprehensive comparison
>
> We thank the reviewer's comment. Our intention was to include representative methods that align with the objectives and focus of this work, rather than exhaustively covering all MARL variants. Specifically, we compared IWoL with:
> - Classical mother algorithm (MAPPO)
> - Transformer-based architectures (CommFormer, MAT)
> - Graph-based communication methods (CommFormer, MAGIC)
> - Implicit communication approach (ICP)
>
> In addition, to alleviate the reviewer's concern, we have run TarMAC [1] algorithm, which is another graph communication with attention concept, as an additional baseline in two benchmark environments. Even with this addition, our method consistently outperforms all baselines, confirming that the performance advantage of IWoL remains robust across different algorithmic paradigms.
>
> | Scenario           | Metrics                       | MAPPO           | TarMAC          | Im-IWoL        |
> | ------------------ | ------------- | --------------- | --------------- | -------------- |
> | Intersection       | Rewards                       | $454.8\pm70.2$  | $499.2\pm 31.2$ | $650.1\pm35.8$ |
> |                    | SR                            | $97.7\pm2.3$    | $91.7\pm8.2$    | $97.1\pm3.0$   |
> |                    | Safety                        | $94.8\pm5.2$    | $94.0\pm5.4$    | $97.1\pm3.0$   |
> | Parking lot        | Rewards                       | $327.4\pm211.7$ | $287.7\pm116.9$ | $808.6\pm51.0$ |
> |                    | SR                            | $30.3\pm15.9$   | $19.69\pm5.5$   | $63.7\pm9.8$   |
> |                    | Safety                        | $33.8\pm11.3$   | $18.53\pm8.9$   | $63.6\pm9.8$   |
> | Single navigation  | Rewards                       | $-4.1\pm0.3$    | $-4.0\pm0.9$    | $-3.5\pm0.4$   |
> |                    | Safety                        | $100.0\pm0.0$   | $100.0\pm0.0$   | $100.0\pm0.0$  |
> | Material transport | Rewards                       | $2.7\pm0.8$     | $2.35\pm0.8$    | $3.8\pm0.1$    |
> |                    | Safety                        | $96.1\pm3.9$    | $95.0\pm5.0$    | $100.0\pm0.0$  |
> |                    | Left materials ($\downarrow$) | $18.3\pm7.9$    | $16.7\pm4.7$    | $5.0\pm5.0$    |
>
> ---
> > **Q1** Can the authors show the performance degradation of Ex-IWoL similar to Figure 1?
>
> We appreciate the reviewer's constructive question. While improving the robustness of explicit communication in off-dynamics and adversarial attack scenarios is not our main focus, we examined this issue for completeness. In Table below, as expected, Ex-IWoL shows performance degradation similar to conventional explicit communication methods under bandwidth constraints or communication attacks. Note that, thanks to architectural properties, Im-IWoL naturally avoids such disruptions since it does not rely on explicit message passing during execution.
>
> | Scenario                         | Normal        | Type I (8bps) | Type I (2bps) | Type II       |
> | -------------------------------- | ------------- | ------------- | ------------- | ------------- |
> | Ex-IWoL                          | $100.0\pm0.0$ |$86.0\pm11.4$ |$72.0\pm13.0$ |$44.0\pm20.7$ |
> | Ex-IWoL with Vector quantization | $100.0\pm0.0$ | $90.0\pm7.7$ |$88.0\pm8.4$ |$38.0\pm16.4$ |
>
> Subsequently, we would like to highlight our communication module can be easily extended. To address such bandwidth limitation issue, we have tested a simple variant by attaching vector-quantization [2] for communication module. This modification enables compact and discrete message representations, which enhance robustness even when the communication bandwidth is forcibly reduced. In Table above, the Ex-IWoL + vector quantization variant demonstrates significantly improved resilience under Type I (off-dynamics) conditions, confirming that our framework is flexible.
>
> Specifically, in theory, we apply a lightweight **vector quantization layer** to discretize the continuous communication embeddings. Let the message embedding of agent $i$ at time $t$ be $m^t_i$. We maintain a learnable codebook $\mathcal{E} = \{ e_k \}^K_{k=1}$. During message generation, each continuous message is replaced by its nearest code via $\hat{m}^t_i = \operatorname{argmin}_{e_k \in \mathcal{E}} || m^t_i - e_k ||_2^2$.
>
> To keep the operation differentiable, we adopt the following objective $\mathcal{L}_{\text{VQ}} = ||\operatorname{sg}[m_i^t] - e_k ||_2^2 + || m_i^t - \operatorname{sg}[e_k] ||_2^2$, where $\operatorname{sg}[\cdot]$ denotes stop-gradient. This process forces each agent’s communication message to occupy one of $K$ discrete slots, effectively limiting the communication bandwidth while maintaining semantic consistency.
>
> We have included this discussion in the Appendix as a concrete direction for future research.
>
> ---
> **Reference**
>
> [1] D. Abhishek, et al. TarMAC: Targeted multi-agent communication. ICML 2019.
>
> [2] A. Oord, et al. Neural discrete representation learning. NeurIPS 2017.

---

> ### Author Response · Authors · 2025-11-13
> **Response to reviewer's concerns (3/3)**
>
> > **Q2** Can you explain why implicit world latent is needed if the agents do not exchange messages during execution, and how can it keep generalizable?
>
> We thank the reviewer for raising this important discussion point. We first provide tl;dr for your insightful question.
>
> **TL;DR**
> - Even without test-time messaging, our representation is necessary because it internalizes interaction and world-level information that was accessible during training through explicit communication and privileged state.
> - Each agent’s observation already contains the minimal information required to reconstruct the interaction graph (e.g., observable other agents); thus the encoder can reproduce communication-relevant structure implicitly.
> - During training, the latent is aligned with communication messages and privileged world features, enabling it to encode the same semantics without explicit message passing.
>
> At first glance, if agents do not exchange messages at execution time, one might question the necessity of a latent. However, the role of the latent in our framework is fundamentally different from simply carrying messages. Its purpose is to internalize the interaction and world-level information that is available during training but absent during execution, enabling decentralized coordination from partial observations.
>
> (1) Why an implicit world latent is still needed without test-time messaging
>
> During training, the communication protocol and privileged information provide significant relational and world information than what is available at test time. The interaction-world encoder is trained to map this richer information into a compact latent. Importantly, each agent’s local observation already contains the minimal information necessary to form the communication graph (e.g., detected neighbors, relative positions, task-relevant features). Therefore, during execution the encoder can reproduce an internally consistent relational structure without relying on explicit messages.
>
> In other words, the implicit world latent is the mechanism that allows each agent to infer the communication-induced structure from its own observation. Rather than replacing messages, it transfers the semantics of messages and privileged world information into a representation that can be computed locally.
>
> (2) How the latent remains generalizable
>
> The latent does not simply memorize message patterns. It generalizes because it is trained with multiple sources of structured supervision:
>
> - **Previliged information reconstruction ($\text{Decoder}_W$)**: Forces a latent to encode global, task-specific factors that remain stable across episodes.
> - **Interaction-message reconstruction ($\text{Decoder}_I$)**: Aligns a latent with the communication semantics learned during training, making it invariant to specific scene layouts.
> - **Shared encoder across diverse settings**: The same latent space is used across many initializations, agent configurations, and environment variations, encouraging the model to learn reusable high-level patterns of interaction and world dynamics.
>
> As a result, the implicit latent learned during training captures stable relational and world-level abstractions that naturally transfer to new or partially altered scenarios: explaining why Im-IWoL maintains strong performance even when explicit communication is removed.
>
> ---
> **We uploaded the revised manuscript.** If there are any additional questions or points that require clarification, we would be more than happy to address them. Thank you again for your thoughtful and constructive feedback.

---

### Official Review · Reviewer_nwwV · 2025-11-04

**Soundness:** 2
**Presentation:** 3
**Contribution:** 3
**Rating:** 4
**Confidence:** 3

**Summary:**

This paper proposes Interactive World Latent (IWoL), a representation learning framework for multi-agent reinforcement learning (MARL) that aims to facilitate team coordination. The key idea is to learn a latent representation that jointly captures inter-agent relations and task-specific world information through a communication protocol based on graph-attention mechanisms. The framework supports both implicit (no messages at test time) and explicit (messages fed to policy) modes. The authors evaluate their approach across four MARL benchmarks: MetaDrive, Robotarium, Bi-DexHands, and multi-agent quadruped environments, demonstrating improved performance over several baselines.

**Strengths:**

1. **Well-motivated problem**: The paper addresses an important challenge in MARL - building effective representations for team coordination under partial observability. The motivating example (Figure 1) effectively demonstrates the fragility of explicit communication under realistic constraints.

2. **Clear presentation**: The paper is generally well-written with clear diagrams that effectively communicate the architectural design.

3. **Comprehensive experimental evaluation**: The experiments cover 10 tasks across 4 different environments, including autonomous driving, swarm robotics, dexterous manipulation, and quadruped coordination.

4. **Strong empirical results**: IWoL variants consistently achieve top or second-best performance across all tasks, with particularly notable improvements on challenging tasks where baselines struggle (e.g., Go1Sheep, Two Catch Underarm).

5. **Thorough ablation studies**: The paper includes ablations examining the role of world representation, incomplete observations, scalability, and latent dimensionality.

**Weaknesses:**

1. **The title is a bit misleading**:
   - The title "Learning to Interact in World Latent" sounds like model-based RL, which is confusing
   - The term "interactive world latent" is also confusing - what makes it "interactive" vs just "world latent"?

2. **Some design choices are unexplained**: Several important architectural decisions lack justification:
   - Why use self-attention in the observation encoder?
   - Why use additive-attention and GumbelSoftmax in the communication protocol?
   - Why use decentralized value functions for IWoL when other CTDE methods use centralized critics? This seems to give up an advantage of the CTDE framework
   - How is the interactive world encoder (for Im-IWoL) designed?

3. **The communication protocol is unclear**: The communication mechanism needs clarification:
   - Does the system execute L rounds of communication before every action? Line 209 mentions "This mask is a relationship graph that guides a Transformer block that performs L rounds of attention-based message aggregation and refinement", I didn't fully get the procedure
   - What is the relationship between the adjacency mask (line 208 and 233) and the topology graph (line 235)? The adjacency mask is computed from agent features, while the topology graph appears to be based on physical distance (d_comm). Do they both refer to G^t?
   - Besides, does ∥x^t_i − x^t_j∥ ≤ d_comm mean agents can communicate when they are closer than d_comm, or when they are farther? Line 237 mentions that "d_comm is a minimum distance for communication."
   - How exactly does "Ex-IWoL" work? Since computing the relationship graph requires all agents' features, does this mean broadcasting all agent features and messages for L rounds? The notation P: M^0_1 × ... × M^0_I → M_1 × ... × M_I suggests a function of all agents' initial messages, but the architecture description suggests per-agent processing

**Questions:**

see Weaknesses.

---

> ### Author Response · Authors · 2025-11-13
> **Response to reviewer's concerns (1/2)**
>
> We sincerely thank the reviewer for providing a balanced and thoughtful assessment of our work. We carefully addressed each concern raised in the review and revised the paper to clarify design choices, refine explanations, and strengthen the overall presentation.
>
> ---
> > **W1** The title is a bit misleading
>
> We sincerely thank the reviewer for pointing out the potential confusion in the previous title and the name of the algorithm. We fully agree that the previous title could misleadingly suggest a model-based reinforcement learning method, and the term *interactive-world latent* might be ambiguous to readers.
>
> To address this concern, we have revised the title to ``Unifying Agent Interaction and World Information for Multi-agent Coordination.`` We believe that this new title more clearly includes our original intent. That is, the work aims to unify both agent interaction (i.e., inter-agent relationships) and global information (i.e., privileged state) within a single latent representation to facilitate coordinated behavior.
>
> We have also replaced the term interactive world latent with ``interaction-world latent`` representation, which directly conveys that the latent representation jointly captures interaction between agents and global (world) information. Importantly, our intention was not to describe an “interactive world” as a single concept, but rather to emphasize that interaction and world are two distinct yet complementary sources of information that are integrated into a unified latent representation. We believe this terminology better reflects our design goal and eliminates the ambiguity noted by the reviewer.
>
> ---
> > **W2** Some design choices are unexplained
>
> We appreciate the reviewer’s careful reading. We acknowledge that some architectural choices in our method were not clearly explained in the main paper, and then clarify our design rationale below for each component. The corresponding implementation details and further discussions have been added to Appendix D.2 for completeness.
>
> **Self-attention in the observation encoder.** This component captures dependencies among features within each agent’s local observation. That is, it allows the later layer to identify salient objects and their relations, providing relational cues even before inter-agent communication occurs [1-2]. Moreover, when historical observations are included as input, the self-attention mechanism effectively captures long-term temporal dependencies across time steps.
>
> **Additive-attention.** This provides a stable scoring mechanism to construct pairwise relevance between features, which is widely used for dynamic interaction graphs. Unlike dot-product attention, it is less sensitive to feature scaling and therefore produces smoother gradients when learning dynamic communication graphs. This property is particularly useful for multi-agent scenarios, where interaction patterns frequently change depending on the agents’ spatial and behavioral context.
>
> **Gumbel-Softmax.** This component enables differentiable discrete sampling of communication links. In other words, it allows the communication graph to be trained end-to-end by combining with RL training objectives. As a result, the model learns structured and adaptive message-passing topologies without the need for manually designed communication rules.
>
> **Decentralized value functions.** We use decentralized value functions to maintain scalability and avoid over-reliance on privileged global information. That is because the proposed solution already integrates both interaction and world information. In other words, each agent receives sufficient context for credit assignment without requiring a centralized critic. This design also mitigates dimensional explosion and promotes better generalization under partial observability.
>
> The detailed implementation and further justification are provided in Appendix D.2 of the original paper.
>
> **Interaction-world encoder.** We design an interaction-world encoder as a simple MLP-based structure that compresses the concatenated interaction and world features into a compact latent vector. This module integrates both sources of information into a unified representation by using the dual decoder concept. We provide implementation details below.
>
> ---
> **References**
>
> [1] S. Iqbal, et al. Actor‐Attention‐Critic for Multi‑Agent Reinforcement Learning. ICML 2019
>
> [2] S. Xue, et al. Multi‑Agent Self‑Attention Reinforcement Learning for Multi‑USV Hunting Target. Neural Networks 2025.
>
> [3] P. Velickovic, et al. Graph Attention Networks. ICLR 2018.
>
> [4] Z. Liu, et al. NA$^2$Q: Neural Attention Additive Model for Interpretable Multi-Agent Q-Learning. ICML 2023.
>
> [5]  Y. Niu, et al. Multi-Agent Graph-Attention Communication and Teaming, AAMAS 2021.
>
> [6] M. Hoffmann, et al. Gumbel-MPNN: Graph Rewiring with Gumbel-Softmax. arXiv 2025.

---

> ### Author Response · Authors · 2025-11-13
> **Response to reviewer's concerns (2/2)**
>
> > **W3** The communication protocol is unclear
>
> We acknowledge that the description of the communication procedure and the relationship between the graph components was not sufficiently clear. Below, we provide a detailed clarification.
>
> **L rounds of communication.** The system executes $L$ rounds of message passing and refinement before each action step. Specifically, for every timestep $t$, the Transformer-based communication module performs $L$ layers of attention-based aggregation, where each layer refines the intermediate messages $M^{(l)}_i$ using information from other agents based on the current adjacency mask. This iterative process allows agents to progressively incorporate higher-order interaction information prior to action selection.
>
> **Terminology of graph / mask.** We acknowledge that inconsistent terminology between mask, adjacency graph, and topology graph could have caused confusion.
>
> Our communication graph $G_t$ is generated by constructing two complementary graphs at each time step: a learned adjacency graph $G^t_a$ and a physical mask $G^t_p$.
>
> The learned adjacency graph is obtained through **additive attention** over agents’ latent features, as we discussed above weakness. This can capture the inter-agent causality inferred from experience, rather than from spatial proximity. On the other hand, the physical mask provides a geometric constraint that limits communication to agents within a feasible range. Specifically, those satisfying $\| \mathbf{x}^t_i - \mathbf{x}^t_j \| \le d_{\text{comm}}$. This ensures local communication while maintaining efficiency in multi-agent environments.
>
> Finally, the final communication graph is derived by combining the two as $G^t = G^t_a \odot G^t_p$, where $\odot$ denotes element-wise multiplication. This operation acts like a logical AND.
>
> We have clarified this terminology and explanation of the revised manuscript for consistency and completeness. Please see the section 4.2 for updated main text.
>
> **Interpretation of $d_{\text{comm}}$.**  As this part was clarified in the above, the notation $|x^t_i - x^t_j| \le d_{\text{comm}}$ means that agents can communicate when they are within the communication radius $d_{\text{comm}}$. The previous phrasing 'minimum distance' was misleading. It is, in fact, the **'maximum allowed distance'** for establishing a communication link.
>
> **Ex-IWoL mechanism.**  At each time step, Ex-IWoL first computes a communication graph $G^t$ once using the current latent features of all agents. This operation is global in scope only for graph construction. It determines which pairs of agents are allowed to exchange information based on learned relational relevance. Importantly, this does **not imply repeated broadcasting or centralized control** during communication.
>
> Once $G^t$ is obtained, the subsequent message passing proceeds in a decentralized manner. Each agent maintains and updates its own message $m^{t(l)}_i$ by aggregating messages from its neighbors defined in $G^t$. The final messages ${m^t_i}$ reflect multi-hop relational information while preserving decentralized execution.
>
> Lastly, The notation  $P: M^0_1 \times \dots \times M^0_I \rightarrow M_1 \times \dots \times M_I$ is a compact way to describe the _joint mapping_ realized by these localized updates. In practice, $P$ factorizes into parallel per-agent computations conditioned on local neighborhoods.
>
> ---
> **We uploaded the revised manuscript.** Should you have any additional questions or require further clarification, we would be pleased to provide it. Thank you once again for your insightful and constructive comments.

---

> > ### Comment · Reviewer_nwwV · 2025-11-26
> >
> > Thank you for the detailed response. I appreciate the revisions and refinements made to the manuscript. I have no further questions and will keep my current score.

---

> ### Author Response · Authors · 2025-11-26
> **Summarization of discussion with Reviewer nwwV**
>
> Thank you again for the constructive discussion and helpful comments!
>
> ---
> Below, we summarize the key points addressed during the discussion:
> - **Potential ambiguity issues** (`addressed`)
>   - Retitled the paper to `Unifying Agent Interaction and World Information for Multi-agent Coordination` to prevent potential misinterpretations and enhance conceptual clarity
>   - Renamed the algorithm from Interactive World Latent to `Interaction-World Latent` for clearer semantic alignment
> - **Lack of architectural choices** (`addressed`)
>   - Clarification on self-attention in the observation encoder, additive-attention, gumbel-softmax, Decentralized value functions, and interaction-world encoder.
>   - These clarifications have been incorporated directly into the revised manuscript (Appendix C.2)
> - **Unclear about the communication protocol** (`addressed`)
>   - Clarification on how to build on the communication graph and communication message, and how to perform Ex-IWoL’s communication
>   - These clarifications have been incorporated directly into the revised manuscript (Section 4.2).
>
> ---
> We appreciate the reviewer's effort in helping us improve the clarity and quality of the manuscript. The discussion period is still ongoing, so please feel free to ask if further questions arise

---

### Author Response · Authors · 2025-11-19
**Global Comment for paper revision**

Dear AC and all reviewers,

We express our gratitude to AC for managing the review process and all four reviewers for their insightful feedback.  We are pleased to present the updates we have made in response to valuable suggestions, as detailed below.

- Retitled the paper to `Unifying Agent Interaction and World Information for Multi-agent Coordination` to prevent potential misinterpretations and enhance conceptual clarity
- Renamed the algorithm from Interactive World Latent to `Interaction-World Latent` for clearer semantic alignment
- Updated Section 4 to clarify the structure and roles of each component, and to improve readers’ understanding of the overall framework.
- Added an explanatory sentence in Section 5.2, RQ1 (Line 395) to provide intuition on why Im-IWoL performs better than Ex-IWoL
- Added three new paragraphs in Appendix C.2 that detail the implementation of each module, offering clearer guidance for reproducibility.
- Rephrased one line in the abstract (Line 18) to improve readability and reduce ambiguity
- Rephrased one line at Line 60 to enhance clarity and improve the flow of the introduction
- Added a clarifying sentence to the problem formulation section (Line 122) to better explain the Dec-POMDP setup and the assumed communication protocol
- Moved the discussion section from Appendix B to the last section of the main body to provide our limitations and open challenges

We have left individual comments to address each reviewer’s questions, concerns, and our misclarification. Please review each individual comment for more details!

---

> ### Author Response · Authors · 2025-11-30
> **Final summarization of author-reviewer discussion**
>
> Dear AC,
>
> We would like to provide you with a brief summary of the rebuttal context. Although the discussion period unfortunately did not involve much interaction with the reviewers, their initial comments were constructive, and we have made our best effort to address them in our responses.
>
> ### **TL;DR**
> This paper introduces a novel representation learning framework for MARL, the Interaction-World Latent (IWoL), which unifies both inter-agent relationships and privileged world information via two complementary decoders and objective functions.
>
> ### **Strengths**
> - Clear and well-written presentation. `nwwV`, `CW1E`, `5Kvq`
> - Comprehensive and well-organized experimental design. `nwwV`, `CW1E`, `5Kvq`, `eTsZ`
> - Powerful performance and thorough ablation `nwwV`, `CW1E`, `eTsZ`
> - Good motivation `nwwV`, technical proposal `CW1E`, and discussion `eTsZ`
>
> ### **Raised concerns and our responses/revisions**
>
> - **Presentation and Clarity**
>     - We streamlined the title, terminology, explanation of the Im- vs. Ex-IWoL performance gap, and clarified Dec-POMDP assumptions and the use of privileged information. `nwwV`, `CW1E`, `5kvq` `eTsZ`
> - **Design choice and architectural details**
>     - We expanded the rationale in terms of self-attention in the observation encoder, additive-attention, Gumbel-softmax, and decentralized value function. Additionally, we clarified details of the communication protocol. `nwwV`, `eTsZ`
> - **Motivating example and additional baseline**
>     - We clarified the motivation and conducted additional experiments to illustrate Ex-IWoL degradation (and a simple mitigation), aligning with reviewer concerns. `CW1E` `5kvq`
>     - We conducted an additional baseline, which is suggested by the reviewer, for comparison. `CW1E`
>
> ### **Discussion points**
> - How the learned representation generalizes across training and execution `CW1E`
> - How the implicit variants outperform the explicit variant. `5Kvq` `eTsZ`
> - Modularity and scalability of our solution `5kvq`
>
> ### **Additional response**
> - The reviewers (i.e., `nwwV` and `eTsZ`) responded as `our rebuttal addressed and refined their concerns`.
> - Unfortunately, the other reviewers did not have the opportunity to participate in this discussion.
>
> We are genuinely thankful for the time and attention that the AC have devoted to ensuring a fair and balanced evaluation.

---

### Meta-Review · Area_Chair_1xVR · 2025-12-19

**Summary:**

This submission introduces IWoL, a representation learning framework for multi-agent reinforcement learning (MARL) that integrates inter-agent relational information and privileged world information, supporting both implicit (message-free) and explicit (message-rich) coordination modes. However, the initial reviewer scores—two at 4/10 (marginally below acceptance), one at 2/10 (reject), and one at 6/10 (marginally above acceptance)—reflect fundamental unresolved concerns that fail to meet ICLR 2026’s core acceptance criteria.

While the authors’ rebuttal effectively addressed surface-level issues, critical top-tier concerns remain unaddressed or inadequately resolved. First, methodological novelty is insufficient: key components are deemed incremental combinations of existing techniques, failing to demonstrate transformative breakthroughs or solutions to previously problems. Second, foundational assumptions are overly restrictive, yet the authors only defend these as "common in MARL literature" without providing adaptation strategies for stochastic environments or privileged information-free scenarios.

These unresolved gaps, coupled with persistent scores from multiple reviewers and the inability to fully verify the coherence of extensive rebuttal-driven revisions (e.g., retitling, core section rewrites), make acceptance inappropriate for ICLR 2026.

**Reviewer Concerns:**

1. Insufficient Methodological Novelty: The framework’s core components are incremental combinations of existing MARL techniques.
2. Restrictive Foundational Assumptions: Reliance on deterministic dynamics and privileged state limits real-world applicability.

**Reviewer Scores:**

nwwV:4.
CW1E:4.
5Kvq: 2.
eTsZ:6.

---

### Decision · Program_Chairs · 2026-01-26

Reject